# Impaired skeletal muscle mitochondrial pyruvate uptake rewires glucose metabolism to drive whole-body leanness

Arpit Sharma[1], Lalita Oonthonpan[1†], Ryan D Sheldon[1†], Adam J Rauckhorst[1], Zhiyong Zhu[2], Sean C Tompkins[1], Kevin Cho[3], Wojciech J Grzesik[4,5], Lawrence R Gray[1], Diego A Scerbo[1], Alvin D Pewa[1], Emily M Cushing[1], Michael C Dyle[2], James E Cox[6,7], Chris Adams[2,4,8,9,10], Brandon S Davies[1,4,9,10], Richard K Shields[4,11], Andrew W Norris[1,4,5,12], Gary Patti[3], Leonid V Zingman[2,4,10,13], Eric B Taylor[1,4,8,9,10,14]*

[1]Department of Biochemistry, Carver College of Medicine, University of Iowa, Iowa City, United States; [2]Department of Internal Medicine, Carver College of Medicine, University of Iowa, Iowa City, United States; [3]Department of Chemistry, School of Medicine, Washington University, St. Louis, United States; [4]Fraternal Order of the Eagles Diabetes Research Center (FOEDRC), Carver College of Medicine, University of Iowa, Iowa City, United States; [5]FOEDRC Metabolic Phenotyping Core Facility, Carver College of Medicine, University of Iowa, Iowa City, United States; [6]Department of Biochemistry, School of Medicine, University of Utah, Salt Lake City, United States; [7]Metabolomics Core Research Facility, School of Medicine, University of Utah, Salt Lake City, United States; [8]Department of Molecular Physiology and Biophysics, Carver College of Medicine, University of Iowa, Iowa City, United States; [9]Pappajohn Biomedical Institute, Carver College of Medicine, University of Iowa, Iowa City, United States; [10]Abboud Cardiovascular Research Center, Carver College of Medicine, University of Iowa, Iowa City, United States; [11]Department of Physical Therapy and Rehabilitation Science, Carver College of Medicine, University of Iowa, Iowa City, United States; [12]Department of Pediatrics, Carver College of Medicine, University of Iowa, Iowa City, United States; [13]Department of Veterans Affairs, Medical Center, Carver College of Medicine, University of Iowa, Iowa City, United States; [14]FOEDRC Metabolomics Core Facility, Carver College of Medicine, University of Iowa, Iowa City, United States

*For correspondence:
eric-taylor@uiowa.edu

†These authors contributed equally to this work

Competing interests: The authors declare that no competing interests exist.

**Abstract** Metabolic cycles are a fundamental element of cellular and organismal function. Among the most critical in higher organisms is the Cori Cycle, the systemic cycling between lactate and glucose. Here, skeletal muscle-specific Mitochondrial Pyruvate Carrier (MPC) deletion in mice diverted pyruvate into circulating lactate. This switch disinhibited muscle fatty acid oxidation and drove Cori Cycling that contributed to increased energy expenditure. Loss of muscle MPC activity led to strikingly decreased adiposity with complete muscle mass and strength retention. Notably, despite decreasing muscle glucose oxidation, muscle MPC disruption increased muscle glucose uptake and whole-body insulin sensitivity. Furthermore, chronic and acute muscle MPC deletion accelerated fat mass loss on a normal diet after high fat diet-induced obesity. Our results illuminate the role of the skeletal muscle MPC as a whole-body carbon flux control point. They highlight the potential utility of modulating muscle pyruvate utilization to ameliorate obesity and type 2 diabetes.

DOI: https://doi.org/10.7554/eLife.45873.001

## Introduction

During type 2 diabetes (T2D) decreased skeletal muscle glucose uptake significantly drives chronic hyperglycemia that leads to microvascular complications and tissue degeneration (*DeFronzo and Tripathy, 2009*). The metabolic and molecular causes of decreased muscle glucose uptake have been intensively investigated but many critical questions remain (*Stanford and Goodyear, 2014*). It is clear that impaired insulin signaling and consequently blunted GLUT4 transporter translocation to the plasma membrane contribute prominently (*Kampmann et al., 2011*). However, the factors underlying muscle insulin resistance are complex, and understanding the relative importance of potential contributing mechanisms has proven challenging.

In addition to hormonal signaling, skeletal muscle glucose disposal is regulated by muscle-autonomous mechanisms. Over shorter time-scales, this is exemplified by the glucose-fatty acid or Randle cycle, where acutely increasing fatty acid supply decreases muscle glucose uptake and oxidation (*Randle et al., 1963*). Similarly, during obesity and T2D chronically elevated muscle lipid content and insulin resistance strongly associate, with increased ceramides, diacylglycerols, and mitochondrial acyl-carnitines and -CoAs implicated as causing insulin resistance (*Koves et al., 2008*; *Samuel and Shulman, 2016*). Thus, acute and chronic lipid oversupply leads to impaired muscle glucose disposal. Notably, a correlate of both conditions is decreased pyruvate dehydrogenase (PDH) activity (*Albers et al., 2015*; *Pilegaard et al., 2006*; *Wu et al., 1999*). PDH converts glucose-derived pyruvate into acetyl-CoA inside mitochondria and is therefore essential for complete glucose oxidation.

In T2D, decreased PDH activity has been hypothesized to impair muscle glucose uptake by metabolic feedback (*Jeoung, 2015*; *Nogiec et al., 2015*). Tests of the converse relationship, whether increasing PDH activity increases glucose uptake do not support this idea and demonstrate that glucose oxidation and uptake are not obligately connected. Acute chemical PDH activation in rats by dichloroacetate infusion increased muscle glucose oxidation but not uptake (*Small et al., 2018*). Furthermore, constitutive genetic PDH activation increased muscle glucose oxidation as expected, but led to lipid accumulation, marked insulin resistance, and impaired glucose uptake (*Rahimi et al., 2014*). Thus, neither chronically nor acutely increasing muscle glucose oxidation increased muscle glucose uptake. Yet, the more directionally relevant question for T2D, how decreasing muscle pyruvate oxidation affects glucose uptake has, to our knowledge, not been selectively tested.

To investigate the effects of decreasing muscle pyruvate oxidation on muscle and systemic glucose metabolism, we generated skeletal muscle-specific mitochondrial pyruvate carrier (MPC) knockout mice (MPC SkmKO). The MPC occupies a central metabolic intersection by transporting cytosolic pyruvate into the mitochondrial matrix, thereby linking glycolysis with oxidative phosphorylation. The genes encoding the MPC were recently discovered, and little is known about how mitochondrial pyruvate uptake affects skeletal muscle function (*Bricker et al., 2012*; *Herzig et al., 2012*). Surprisingly, MPC disruption did not impair muscle glucose uptake. Instead, it evoked complex, interconnected changes in muscle and systemic metabolism leading to increased whole-body insulin sensitivity, increased muscle glucose uptake, and attenuation of obesity and T2D.

## Results

### Skeletal muscle-specific MPC deletion abolishes mitochondrial pyruvate uptake and increases whole-body energy expenditure

The mammalian Mitochondrial Pyruvate Carrier (MPC) comprises two obligate, paralogous subunits, MPC1 and MPC2. Loss of either subunit results in destabilization and loss of the other subunit and thus the MPC complex. We generated skeletal muscle-specific MPC knockout mice (MPC SkmKO) by crossing mice with a floxed *Mpc1* allele (*Mpc1$^{fl}$*) (*Gray et al., 2015*) with mice expressing Cre under control of a *Myogenin* promoter (*Myogenin*-Cre) (*Figure 1A*) (*Fink et al., 2018*; *Li et al., 2005*). *Myogenin*-Cre selectively recombines floxed alleles in skeletal muscle. Western blots confirmed selective loss of MPC1and MPC2 proteins in MPC SkmKO mouse muscles (*Figure 1B*). *Mpc1* but not *Mpc2* mRNA was also lost (*Figure 1—figure supplement 1*). In assays with isolated skeletal

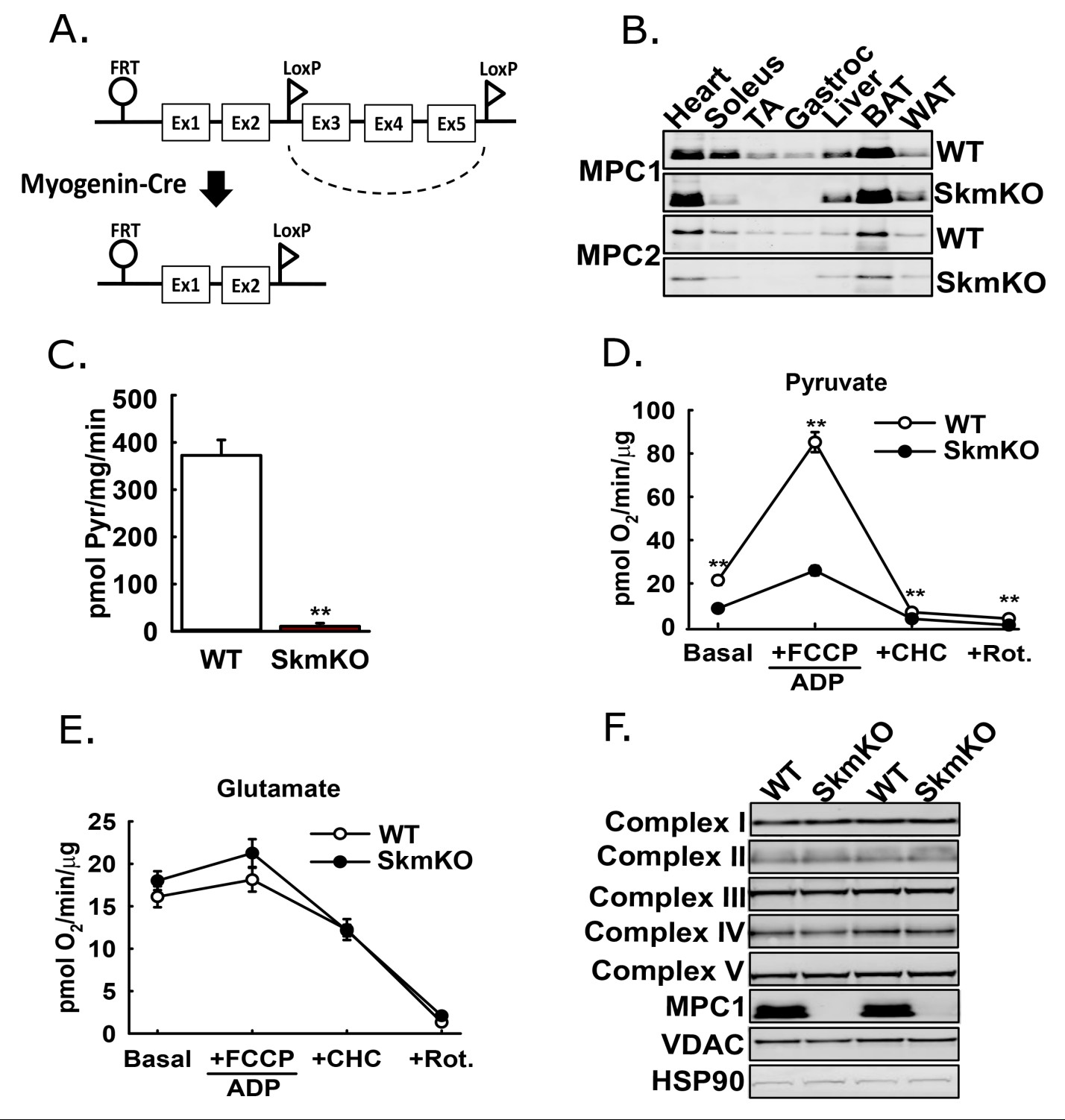

**Figure 1.** Generation of mice with muscle-specific deletion of *Mpc1* (MPC SkmKO). (A) Scheme illustrating generation of the muscle-specific *Mpc1* null allele. (B) Representative western blots of MPC1 and MPC2 protein abundance in mouse tissues. Loading was normalized to total protein. A reference protein is not shown because of lack of an equally expressed protein across different tissues (age 21 weeks; n = 3, littermates; TA, tibialis anterior; BAT, brown adipose tissue; WAT, white adipose tissue). (C) $^{14}C$-pyruvate uptake by muscle mitochondria isolated from WT and MPC SkmKO mice (age 13 weeks, n = 6, four littermates + 2 non-littermates, two-tailed t-test). (D, E) Pyruvate- (D) and glutamate-driven (E) respiration by muscle mitochondria isolated from WT and MPC SkmKO mice. Experimental media contained 1 mM malate and 10 mM pyruvate or 10 mM glutamate (age 16 weeks; n = 6, four littermates + 2 non-littermates; two-tailed t-test; FCCP, trifluoromethoxy carbonylcyanide phenylhydrazone; CHC, 4-alpha-hydroxycinnamatic acid;
*Figure 1 continued on next page*

*Figure 1 continued*

Rot., rotenone). (F) Representative western blots of components of electron transport chain (ETC) complexes I-V, MPC1, VDAC, and HSP90 proteins in TA muscle from WT and MPC SkmKO mice (age 21 weeks, n = 6, littermates). Data presented as mean ± SEM (*p<0.05, **p<0.01).

DOI: https://doi.org/10.7554/eLife.45873.002

The following figure supplement is available for figure 1:

**Figure supplement 1.** Loss of *Mpc1* but not *Mpc2* transcript in MPC skmKO skeletal muscle.

DOI: https://doi.org/10.7554/eLife.45873.003

muscle mitochondria, MPC disruption abolished $^{14}$C-labeled pyruvate uptake and markedly decreased pyruvate-driven respiration (*Figure 1C and D*). In contrast, glutamate oxidation and markers of electron transport chain complexes were not different (*Figure 1E and F*). These results are consistent with MPC disruption causing a selective defect in mitochondrial pyruvate metabolism.

To test how decreased mitochondrial pyruvate uptake affects muscle function, we examined muscle strength and endurance. WT and MPC SkmKO mouse grip strength was not different nor was ex vivo muscle force production (*Figure 2A*, *Figure 2—figure supplement 1*). In contrast, MPC SkmKO mice fatigued earlier during treadmill running exercise tolerance tests (*Figure 2B*). In home cage living, MPC SkmKO mice exhibited similar food intake, mildly elevated $VO_2$, decreased RER, and unchanged voluntary locomotion (*Figure 2C, D, E and F*). Muscle glycogen content as a measure of muscle energy balance was also unchanged (*Figure 2—figure supplement 1*). These results indicate that basal MPC SkmKO muscle function is maintained by adaptive mechanisms that increase metabolic rate but cannot support the greater energetic demands of exercise.

## MPC SkmKO mice gain less fat with age and have increased muscle fatty acid oxidation

Because muscles oxidize amino acids as an alternative substrate to glucose, we considered whether impairing muscle pyruvate oxidation would lead to catabolic muscle mass loss. By body weight and composition, WT and MPC SkmKO mice are initially indistinguishable. Surprisingly, beginning about 13 weeks of age, MPC SkmKO mice display a striking resistance to fat mass gains, with perfect lean mass sparing (*Figure 3A and B*). MPC SkmKO mouse leanness and normal grip strength persists through 40 weeks of age (*Figure 3—figure supplement 1*). Interestingly, serum FGF21 was not increased in MPC SkmKO mice (*Figure 3—figure supplement 1*). This is notable because it contrasts with other mouse models of muscle mitochondrial stress, where dramatically increased FGF21 leads to decreased adiposity (*Pereira et al., 2017*; *Tezze et al., 2017*; *Voigt et al., 2015*). Although male mice were used throughout this study, we followed a cohort of female WT and MPC SkmKO mice for 28 weeks and observed similar MPC SkmKO mouse leanness (*Figure 3—figure supplement 2*). In an important control experiment, *Myogenin* Cre expression in *Mpc1*$^{+/+}$ mice did not affect body composition, indicating *Mpc1* deletion but not the presence of Cre causes MPC SkmKO leanness (*Figure 3—figure supplement 2*).

Our observation that MPC SkmKO mice had a lower RER and were leaner suggested muscle fatty acid oxidation could be increased. To test this, we administered [9-10]-$^3$H triolein to mice by retroorbital injection. Relative $^3$H partitioning into the aqueous versus organic fraction, which increases with fatty chain shortening, was greater in MPC SkmKO muscle, indicating increased fatty acid uptake and oxidation (*Figure 3C*). We then tested for muscle autonomous effects by measuring fatty acid oxidation ex vivo in incubated extensor digitorum longus (EDL) muscles. MPC SkmKO muscle manifested greater complete and partial palmitate oxidation (*Figure 3D*).

## Skeletal muscle MPC disruption leads to increased muscle glucose uptake, lactate excursion, and Cori Cycling

Given the essential role of pyruvate oxidation in complete glucose disposal, we examined how muscle MPC disruption affects whole-body glucose metabolism. Glucose tolerance tests (GTTs) at age 7 weeks, before onset of MPC SkmKO mouse leanness, revealed a minimal decrease in MPC SkmKO mouse glucose tolerance, with slightly increased lactate excursion (*Figure 3E and F*). In contrast, at age 36 weeks, well after onset of MPC SkmKO mouse leanness, glucose tolerance was mildly improved and lactate excursion was markedly increased (*Figure 3G and H*). Similar results were

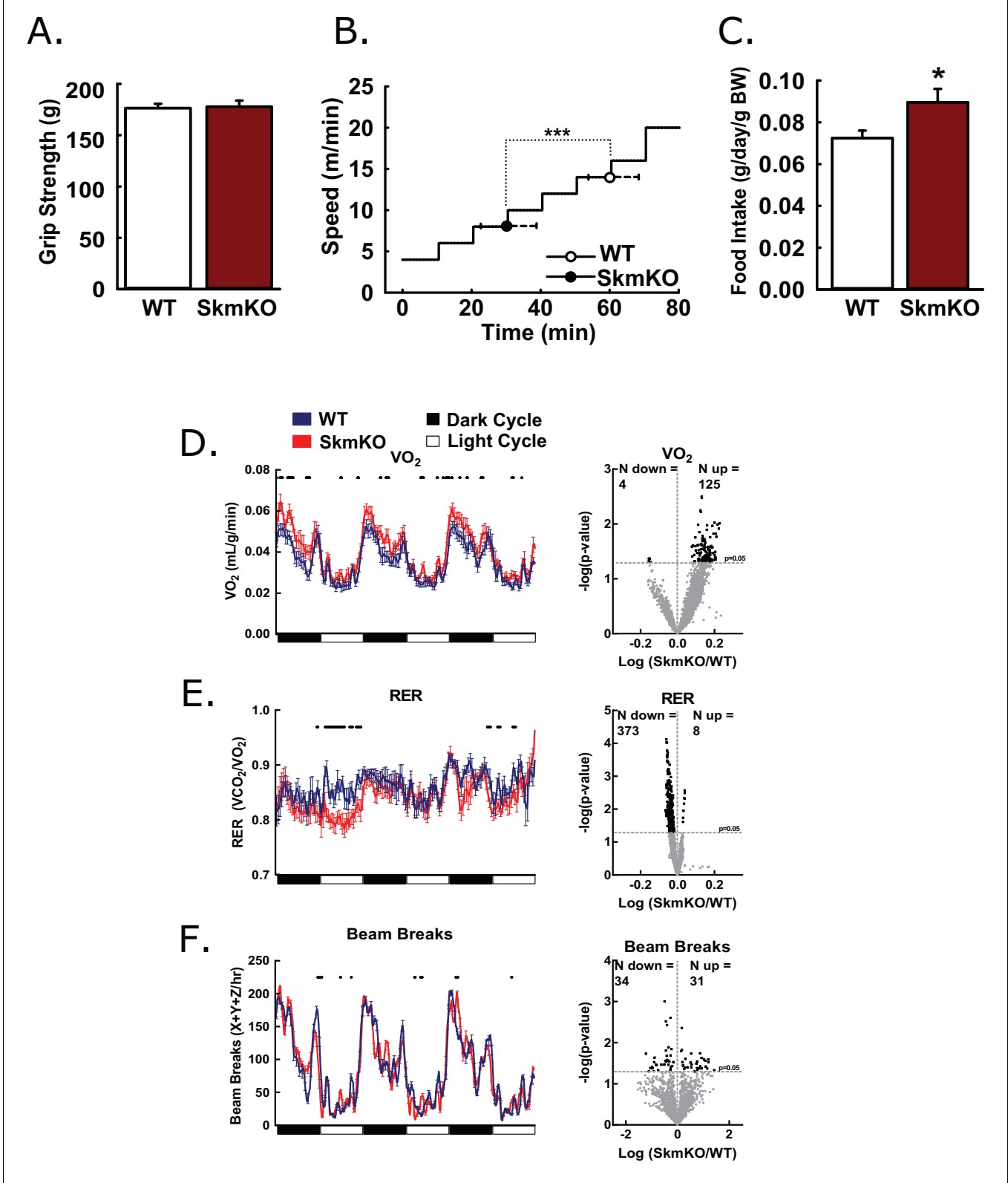

**Figure 2.** Basic characterization of MPC SkmKO mice. (**A**) Grip strength measured by full force on a triangular bar of WT and MPC SkmKO mice (age 15 weeks, n = 8, seven littermates + 1 non-littermate, two-tailed t-test). (**B**) Exercise tolerance of WT and MPC SkmKO mice measured by running duration and speed at exhaustion on a rodent treadmill where belt velocity was incrementally increased (age 14 weeks, n = 6, littermates, two-tailed t-test). (**C**) Body weight (BW)-normalized daily food intake of WT and MPC SkmKO mice (age 15 weeks, n = 8, seven littermates + 1 non-littermate, two-

*Figure 2 continued on next page*

*Figure 2 continued*

tailed t-test). (D - F) Oxygen consumption (VO$_2$) (D), Respiratory exchange ratio (RER) (E), and voluntary locomotion (beam breaks) (F) of WT and MPC SkmKO mice (age 12 weeks, n = 8, littermates, 60 min rolling averages analyzed by two-tailed t-test). Black dots indicate points where significant differences were detected and volcano plots show the distribution of p-values plotted by the direction of change. Data presented as mean ± SEM (*p<0.05 and as indicated on volcano plots).
DOI: https://doi.org/10.7554/eLife.45873.004
The following figure supplement is available for figure 2:

**Figure supplement 1.** Skeletal muscle specific force production and glycogen content are unchanged in MPC SkmKO mice.
DOI: https://doi.org/10.7554/eLife.45873.005

observed during a fasting and refeeding experiment performed with a separate cohort of mice at age 14 weeks, shortly after MPC SkmKO mouse leanness emerged (*Figure 3—source data 1*).

We considered whether changes in glucose metabolism resulting from muscle MPC disruption were greater than what was detected by GTTs, because of the potential for increased muscle lactate excretion to drive increased hepatic gluconeogenesis. If so, this could mask increased glucose disposal in MPC SkmKO mice during GTTs. To assess whole-body insulin sensitivity and glucose metabolism, we performed traced, hyperinsulinemic-euglycemic clamps (*Figure 4A*). Glucose infusion rate (GIR) was moderately greater in MPC SkmKO mice during steady state, indicating greater whole-body insulin sensitivity (*Figure 4B*). Under basal conditions, MPC SkmKO mice showed increased hepatic glucose production (Ra) and peripheral disposal (Rd), indicating elevated glucose turnover, or Cori Cycling (*Figure 4C and D*). In contrast, under clamped steady state conditions, when insulin is increased by exogenous administration (*Figure 4—figure supplement 1*), MPC SkmKO mice showed no difference in Ra but significantly greater Rd (*Figure 4C and D*). Notably, clamped steady state MPC SkmKO muscle glucose uptake and blood lactate levels were strikingly increased (*Figure 4E and F*). Western blots of mixed quadriceps muscle harvested at the end of clamps did not show increased MPC SkmKO muscle Akt serine 473, AMPK threonine 172 phosphorylation, or PDH E1α serine 232 phosphorylation, indicating that if these regulatory protein modifications contributed to increased MPC SkmKO glucose uptake that they did not persist until time of analysis (*Figure 4—figure supplement 2*).

The combined data from these clamp experiments are consistent with two gluconeogenesis control tiers, mass action and hormonal. Under basal conditions, constitutively elevated muscle lactate excretion in MPC SkmKO mice drives increased hepatic gluconeogenesis and Cori Cycling. Under clamped steady state conditions, insulin administration supersedes mass action and suppresses gluconeogenesis proportionately greater in MPC SkmKO mice to equal that of WT mice. Thus, muscle MPC disruption moderately increased metabolic flexibility of the liver.

## In situ muscle contraction reveals mechanisms of metabolomic adaptation

We performed experiments to understand metabolomic adaptations to muscle MPC disruption and the more global MPC role in the muscle metabolic network. We reasoned that muscle contraction would amplify differences between WT and MPC SkmKO muscle by increasing energy demand. We contracted the tibialis anterior (TA) muscle of live, anesthetized mice in situ by electrically stimulating the peroneal nerve, with sham treatment of the other limb as the non-contracted control. To detect metabolomic changes resulting from MPC disruption rather than force production differences, we identified a contraction protocol eliciting similar WT and MPC SkmKO muscle fatigue. Compared to WT muscle, MPC SkmKO muscle fatigued more rapidly when stimulated with one twitch per second (1 Hz) but similarly when stimulated with one twitch per two seconds (0.5 Hz) (*Figure 5A*, *Figure 5—figure supplement 1*). For metabolomic analysis, sham and 0.5 Hz contracted muscles were harvested immediately after treatment and clamp frozen at liquid nitrogen temperature. Muscle extracts were analyzed by mass spectrometry to compare levels of 62 metabolites (*Supplementary file 1*). Principal component analysis demonstrated distinct effects of both contraction and genotype (*Figure 5—figure supplement 1*).

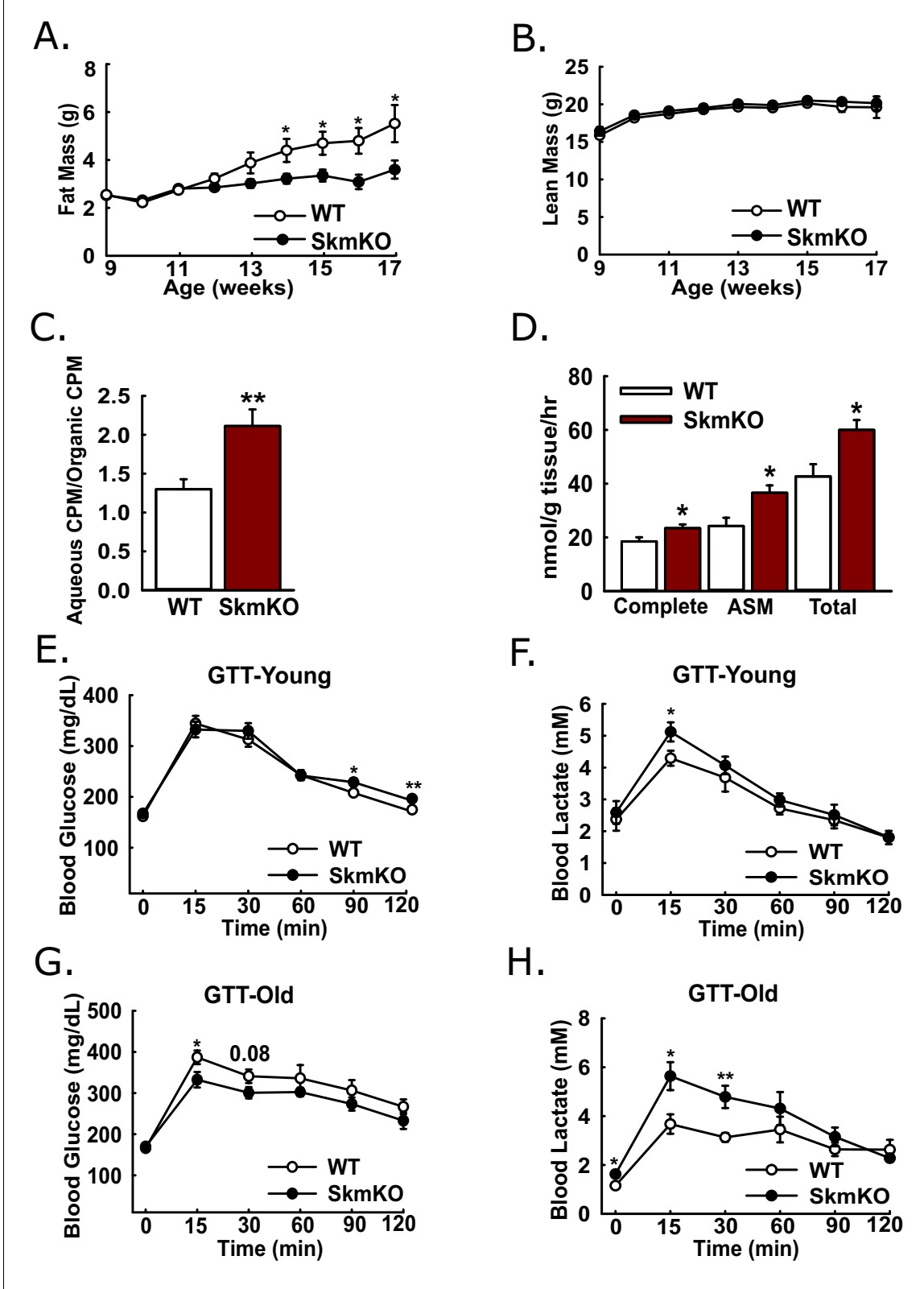

**Figure 3.** Leanness, increased fatty acid oxidation, and altered systemic glucose metabolism. (A-B) Total fat (A) and lean (B) mass of live WT and MPC SkmKO mice measured by NMR (ages 9–17 weeks, n = 6 littermates, two-tailed t-test). (C) Ratio of muscle $^3$H-triolein uptake and partitioning into aqueous and organic fractions (age 16 weeks, n = 7, littermates, two-tailed t-test). (D) Oxidation of $^{14}$C-palmitate by incubated EDL muscles to $CO_2$ (complete) and acid soluble metabolites (ASM) (age 13 weeks, before MPC SkmKO leanness, n = 6, littermates, two-tailed t-test; EDL, extensor

*Figure 3 continued on next page*

*Figure 3 continued*

digitorum longus). (E, F) Blood glucose (E) and lactate (F) levels during glucose tolerance tests with 7 week-old (young) WT and MPC SkmKO mice (n=17, 15 littermates, 2 non-littermates, two-tailed t-test). (G, H) Blood glucose (G) and lactate (H) levels during glucose tolerance tests with 36 week-old (old) WT and MPC SkmKO mice (n=8, littermates, two-tailed t-test). Data presented as mean ± SEM (*p<0.05, **p<0.01).

DOI: https://doi.org/10.7554/eLife.45873.006

The following source data and figure supplements are available for figure 3:

**Source data 1.** Serum parameters of 12 hr fasted and 30 min post refed WT and MPC SkmKO mice.
DOI: https://doi.org/10.7554/eLife.45873.009

**Figure supplement 1.** Fat mass loss and lean mass retention in MPC SkmKO mice.
DOI: https://doi.org/10.7554/eLife.45873.007

**Figure supplement 2.** Fat mass loss in female MPC SkmKO mice and not Myogenin-CRE/+ mice.
DOI: https://doi.org/10.7554/eLife.45873.008

## MPC disruption decreases muscle pyruvate oxidation and transfers reductive drive to the liver

MPC SkmKO muscle exhibited metabolomic changes consistent with impaired mitochondrial pyruvate utilization. Compared to contracted WT muscle, contracted MPC SkmKO muscle pyruvate levels were several fold greater; conversely, lactate was similarly increased in contracted WT and SkmKO muscle (*Figure 5B*). These results suggest contraction increases glycolytic production of pyruvate that is readily oxidized by WT but not SkmKO muscle. Pyruvate and lactate rapidly equilibrate across lactate dehydrogenase utilizing NADH and $NAD^+$ as co-factors. Accordingly, the lactate:pyruvate ratio, which was markedly greater in contracted WT compared to MPC SkmKO muscle, is often reflected by the NADH:$NAD^+$ ratio. This is precisely what we observed here, where $NAD^+$ was unchanged and NADH was decreased in MPC SkmKO muscle during contraction (*Figure 5B*). Of note, in muscle the NADH pool is several fold greater than the $NAD^+$ pool (*Steen and Vesth, 1979*). Thus, the decrease in NADH does not necessarily indicate a decrease in the combined NADH + $NAD^+$ pool. MPC disruption decreased isocitrate/citrate and malate, and abolished contraction-induced increases in succinate (*Figure 5C*). In contrast, no genotype effect was observed for α-keto-glutarate (αKG), which may be derived from glutamate independently of pyruvate fluxes (*Figure 5C*). Together, the decreased lactate:pyruvate ratio, NADH:$NAD^+$ ratio, and TCA cycle metabolite levels in SkmKO muscle convincingly demonstrate a shift to a more oxidized state. This is consistent with impaired capacity to generate reducing equivalents from pyruvate, resulting in transfer of reductive drive to the liver in the form of lactate.

## MPC disruption increases acyl-carnitine utilization

We next considered how muscle MPC disruption could affect acyl-carnitine content. Muscle acyl-carnitine accumulation and mitochondrial overload is associated with insulin resistance in T2D (*Koves et al., 2008*). Thus, compensatory MPC SkmKO fatty acid oxidation could protect from insulin resistance by increasing acyl-carnitine clearance. Indeed, MPC disruption strikingly attenuated contraction-induced increases in acyl-carnitine content (*Figure 5D*). No main effects of genotype were observed for carnitine levels, indicating free carnitine was not a limiting substrate for acyl-carnitine production (*Figure 5D*). Lower acyl-carnitine levels in contracted MPC SkmKO muscle could result from either decreased production or increased oxidation. Our observations of increased fatty acid oxidation in MPC SkmKO muscle suggest the latter.

Enzyme expression changes, increased fatty acid oxidation enzyme specific activities, and diminished TCA cycle substrate competition could each contribute to increased MPC SkmKO muscle fatty acid oxidation. qPCR measurements showed no differences between WT and MPC SkmKO mice for the key fatty acid oxidation enzyme transcripts *Cpt1b*, *Echs1*, and *Hadha* (*Figure 5—figure supplement 2*). As expected, control qPCR measurements for *Mpc1* and *Mpc2* showed near total loss and no difference in MPC SkmKO vs WT muscle, respectively (*Figure 5—figure supplement 2*). These results indicate programmatic upregulation of fatty acid oxidation gene expression does not underlie increased MPC SkmKO muscle fatty acid oxidation.

We performed in vivo universally $^{13}$C-labeled glucose [(U)$^{13}$C-glucose] tracing experiments in resting, 6 hour-fasted mice to understand metabolic mechanisms for increased MPC SkmKO muscle

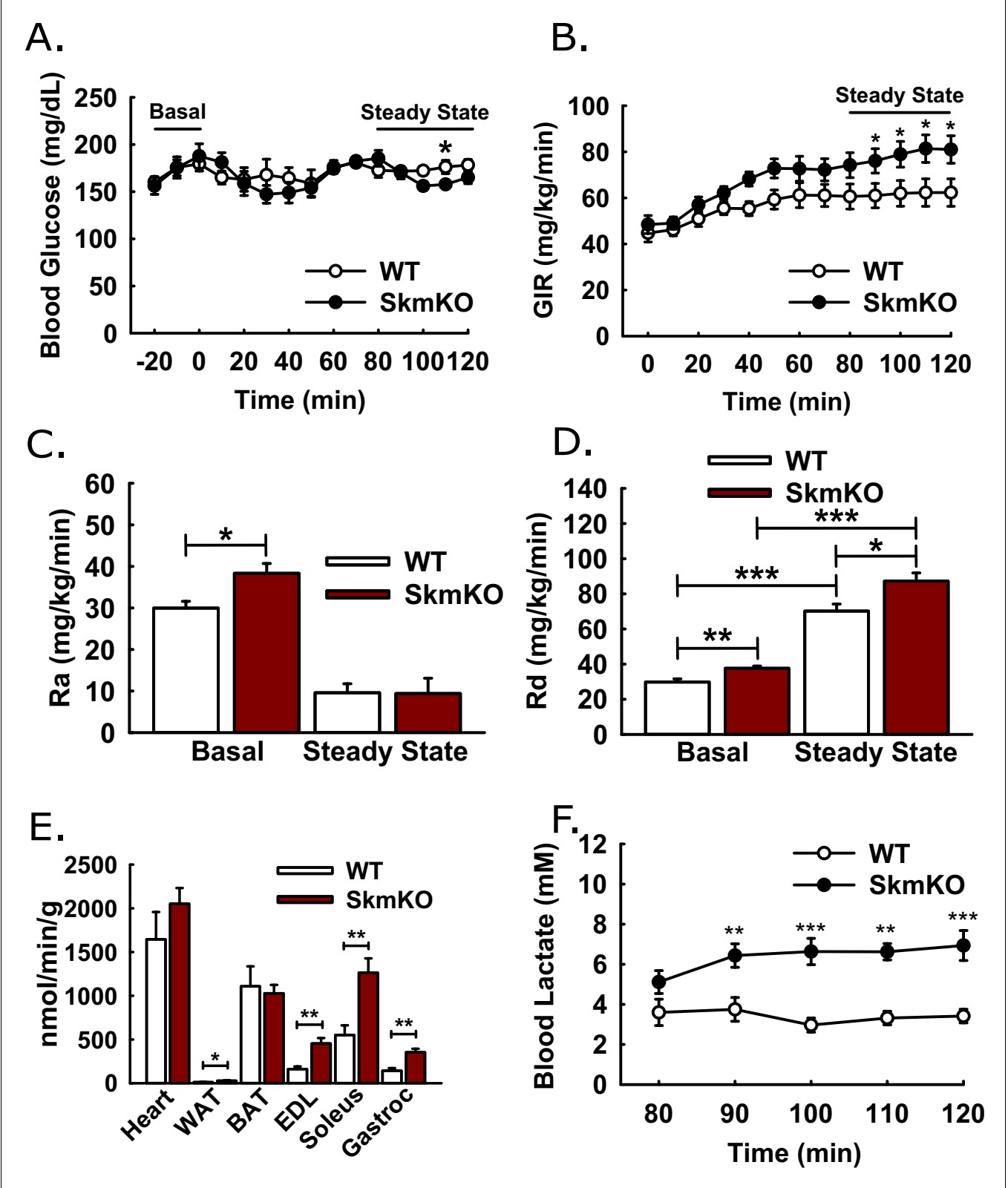

**Figure 4.** Hyperinsulinemic-euglycemic clamps. Clamps were performed on 5 hr fasted, unrestrained, and conscious WT and MPC SkmKO mice. After the basal sampling period, whole body glucose flux was traced by infusion of 2.5 mU/kg/min insulin and of D-[3 $^{H3}$]-glucose at time t = 0. At 45 min prior to clamp conclusion, [1-$^{14}$C]−2-deoxy-D-glucose was infused over 5 min in a single bolus. Tissue samples were collected at clamp conclusion (ages 24–29 weeks, n = 6–8, six littermates + 2 non-littermates, two tailed t-test). (A - D) Blood glucose levels (A), glucose infusion rate (GIR) (B),

*Figure 4 continued on next page*

Figure 4 continued

appearance rate (Ra) (C), and disposal rate (Rd) (D) (ages 24–29 weeks, n = 6–8, six littermates + 2 non-littermates, two tailed t-test). (E, F) Tissue [1-$^{14}$C]−2-deoxy-D-glucose uptake (E) and blood lactate levels (F) during the steady-state portion of the clamp (ages 24–29 weeks; n = 6–8; six littermates + 2 non-littermates; two tailed t-test; WAT, white adipose tissue, BAT, brown adipose tissue, EDL, extensor digitorum longus, Gastroc, gastrocnemius). Data presented as mean ± SEM (*p<0.05, **p<0.01).
DOI: https://doi.org/10.7554/eLife.45873.010

The following figure supplements are available for figure 4:

**Figure supplement 1.** Basal and steady state plasma insulin concentrations during hyperinsulinemic-euglycemic clamps.
DOI: https://doi.org/10.7554/eLife.45873.011

**Figure supplement 2.** AKT, AMPK, and PDH phosphorylation in gastrocnemius muscle collected after hyper-insulinemic euglycemic clamps.
DOI: https://doi.org/10.7554/eLife.45873.012

fatty acid oxidation. In addition to our observation that contracted MPC SkmKO muscle citrate levels were decreased, we observed decreased flux of intraperitoneally-injected (U)$^{13}$C-glucose into muscle citrate, simultaneous with increased flux of $^{13}$C into the liver citrate (*Figure 5E*, *Figure 5—figure supplement 3*, *Supplementary file 2*). The combination of decreased and increased glucose enrichments into muscle and liver TCA cycle intermediates, respectively, is consistent with the elevated Cori Cycling we observed during glucose clamps. Decreased muscle pyruvate flux through citrate could also propagate to decreased malonyl-CoA production and a regulated fatty acid oxidation increase by Cpt1b disinhibition. In a separate experiment under similar conditions but without glucose injection, MPC SkmKO muscle malonyl-CoA levels were decreased (*Figure 5F*). Notably, these TCA cycle tracing and malonyl-CoA quantitation experiments were performed with resting muscle. Conversely, during in situ contraction experiments the greater WT muscle acyl-carnitine increase indicates Cpt1b activity was not limiting WT fatty acid oxidation. In accord, the minimal MPC SkmKO muscle increase in acetyl-carnitine, which is chemically adjacent to acetyl-CoA, is consistent with diminished substrate competition from mitochondrial pyruvate for entry into the mitochondrial acetyl-CoA pool. These observations suggest decreased TCA cycle pyruvate entry in MPC SkmKO muscle leads to increased fatty acid oxidation by both disinhibition of Cpt1b and diminished substrate competition, with the latter predominating in working muscle.

## Adaptive glutaminolysis and pyruvate-alanine cycling

Increased fatty acid oxidation can only partially compensate for decreased pyruvate oxidation, because it requires but does not generate oxaloacetate. Thus, we expected to also observe changes in muscle amino acid content consistent with adaptive mitochondrial utilization. In contrast to the accumulation of pyruvate we observed in MPC SkmKO muscle with contraction, glutamine and glutamate were depleted (*Figure 5G*). Glutamine is a major cellular fuel, and our results suggest that MPC disruption increases glutaminolysis in skeletal muscle as previously observed in cancer cell and liver systems (*Gray et al., 2015*; *Vacanti et al., 2014*; *Yang et al., 2014*). Significant changes were observed in several additional amino acids (*Supplementary file 1*). Two are notable here because of their potential relationship to adaptive MPC SkmKO muscle pyruvate utilization. Alanine was increased similarly by contraction in both WT and MPC SkmKO muscle whereas aspartate was markedly increased in SkmKO muscle only (*Figure 5H*). Pyruvate-alanine cycling can enable MPC-independent pyruvate carboxylation, which could increase aspartate by mitochondrial transamination of oxaloacetate.

We and others previously found that pyruvate-alanine cycling enables MPC-independent mitochondrial pyruvate utilization in fibroblasts and liver (*Bowman et al., 2016*; *Gray et al., 2015*; *McCommis et al., 2015*). In this case, pyruvate is transaminated in the cytosol to alanine, imported into the mitochondrial matrix by a currently unknown alanine transporter, and then deaminated back to pyruvate for oxidation. Here, in addition to our observation that (U)$^{13}$C-glucose flux into citrate was decreased in MPC SkmKO muscle, we observed that (U)$^{13}$C-glucose flux into alanine was increased in MPC SkmKO muscle, consistent with pyruvate-alanine cycling (*Supplementary file 2*). To test for pyruvate-alanine cycling, we incubated permeabilized mouse soleus muscle with the MPC inhibitor UK-5099 and alanine transaminase inhibitor β-chloro-alanine, in both orders of administration (*Figure 5I*). MPC and alanine transaminase inhibition additively decreased pyruvate-driven respiration. To account for the possibility that UK-5099 and β-chloro-alanine were impairing electron

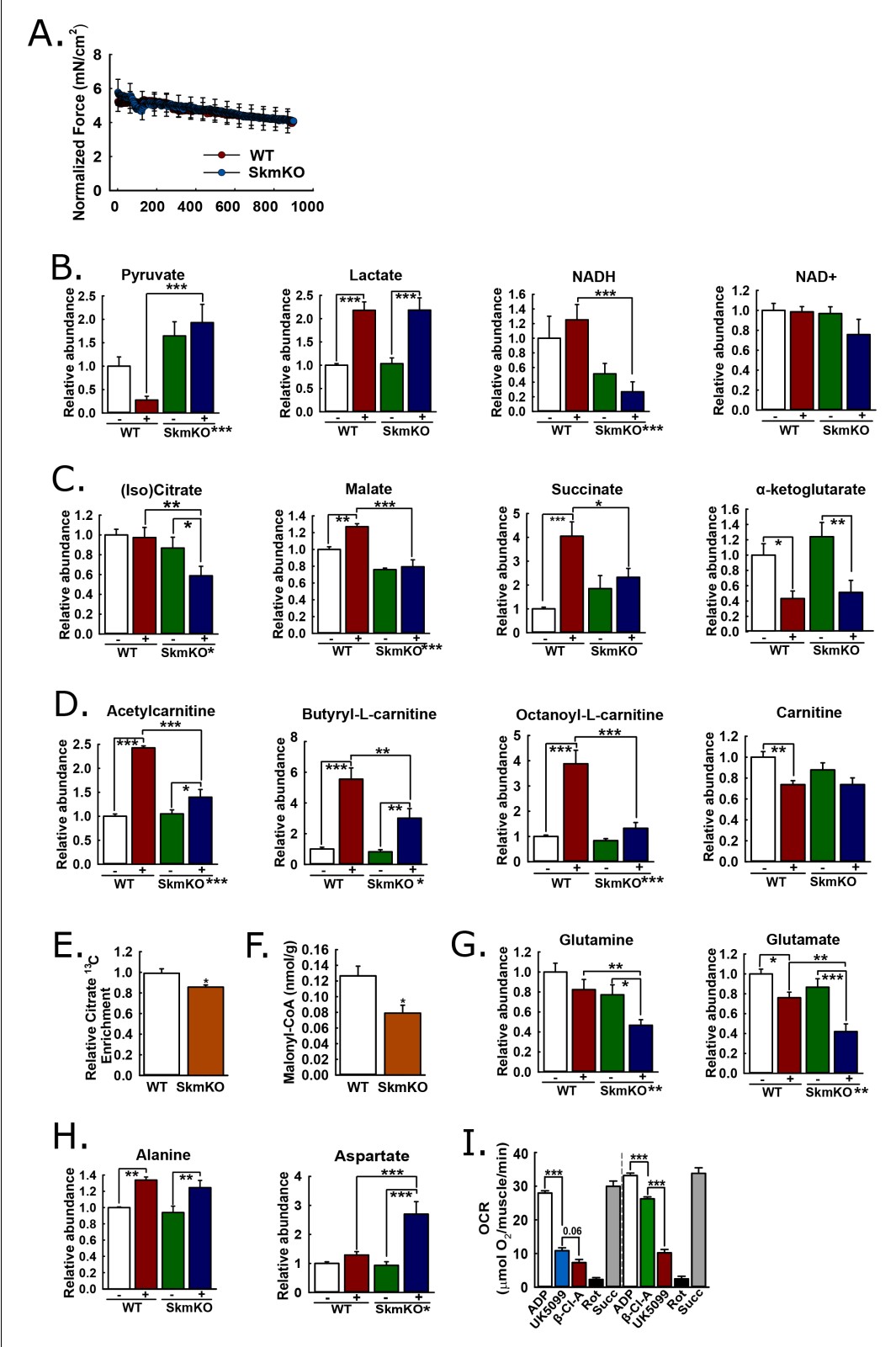

**Figure 5.** Mechanisms of metabolic adaptation. (**A**) Tibialis anterior (TA) muscle force production during 0.5 Hz isometric contraction by in situ peroneal nerve stimulation of live anesthetized WT and MPC SkmKO mice (age 12 weeks, n = 5–6, littermates, 30 s rolling averages analyzed by two-tailed t-test). (**B**) Relative metabolite abundance of pyruvate, lactate, NADH, and NAD$^+$ in sham-treated (-) and contracted (+) TA muscles of WT and MPC SkmKO mice (age 12 weeks, n = 5–6, littermates, two-way ANOVA). (**C**) Relative metabolite abundance of isocitrate +citrate (Iso)Citrate, malate,

*Figure 5 continued on next page*

*Figure 5 continued*

succinate, and α-ketoglutarate in sham-treated (-) and contracted (+) TA muscles of WT and MPC SkmKO mice (age 12 weeks, n = 5–6, littermates, two-way ANOVA). (D) Relative metabolite abundance of acetylcarnitine, butyryl-L-carnitine, octanoyl-L-carnitine, and carnitine in sham-treated (-) and contracted (+) TA muscles of WT and MPC SkmKO mice (age 12 weeks, n = 5–6, littermates, two-way ANOVA). (E) Citrate $^{13}$C enrichment ($^{13}$C labeled/non-labeled citrate) in TA muscles 65 min after intraperitoneal U$^{13}$C-glucose injection of WT and MPC SkmKO mice (age 12 weeks, n = 8, seven littermates + 1 non-littermate, two-way ANOVA). (F) Malonyl-CoA levels in quadriceps muscles of WT and MPC SkmKO mice (age 11 weeks, n = 6, littermates, two tailed student's t test). (G-H) Relative metabolite abundance of glutamate and glutamine (G) and alanine and aspartate (H) in sham-treated (-) and contracted (+) TA muscles of WT and MPC SkmKO mice (age 12 weeks, n = 5–6, littermates, two-way ANOVA). (I) Ex vivo pyruvate-driven, ADP-stimulated respiration of permeabilized mouse soleus muscle treated with UK5099, β-chloro-alanine (β-Cl-A), rotenone (Rot), and rescued with succinate (age 11 weeks, n = 6, littermates, two-tailed t-test). Data are presented as mean ± SEM (*p<0.05, **p<0.01, ***p<0.001, placement on x-axis signifies genotype main effect).

DOI: https://doi.org/10.7554/eLife.45873.013

The following figure supplements are available for figure 5:

**Figure supplement 1.** TA muscle force production during 1 Hz in situ contraction and metabolomics principal component (PCA) analysis of 0.5 Hz contracted TA muscles after in situ contraction.

DOI: https://doi.org/10.7554/eLife.45873.014

**Figure supplement 2.** Transcript abundance of key fatty acid oxidation enzymes is not changed in MPC SkmKO skeletal muscle.

DOI: https://doi.org/10.7554/eLife.45873.015

**Figure supplement 3.** Increased liver citrate $^{13}$C enrichment in MPC SkmKO mice following intraperitoneal injection of U$^{13}$C-glucose.

DOI: https://doi.org/10.7554/eLife.45873.016

transport chain function, we rescued respiration by treatment with the Complex I inhibitor rotenone, to prevent reverse electron transport, and the Complex II substrate succinate. Succinate restored respiration to pre-inhibited levels, indicating maintenance of gross mitochondrial oxidative capacity. These results provide evidence for pyruvate-alanine cycling as a mechanism for MPC-independent carbohydrate oxidation in MPC SkmKO muscle.

Combined with our observations of decreased MPC SkmKO muscle glucose flux into citrate, our data also suggest that pyruvate-alanine cycling is kinetically limited compared to direct mitochondrial pyruvate uptake. During in vivo (U)$^{13}$C-glucose tracing, the citrate M + 1 and M + 3 isotopologues were decreased in MPC SkmKO muscle, indicating more greatly decreased pyruvate anaplerosis through pyruvate carboxylase compared to forward TCA cycle flux through PDH (*Supplementary file 2*). Thus, pyruvate-alanine cycling may be more specifically limiting for pyruvate flux through pyruvate carboxylase and require concomitantly increased glutamine anaplerosis for maintaining TCA cycle flux. This is consistent with the decreased MPC SkmKO but not WT muscle glutamine levels we observed during in situ contraction. Together with observed fatty acid oxidation increases, these data support a model where increased acyl-carnitine utilization, pyruvate-alanine cycling, and anaplerotic glutaminolysis sustain TCA cycle activity when direct pyruvate transport is lost by MPC disruption.

## Constitutive and acute skeletal muscle MPC deletion enhances recovery from high fat diet-induced obesity

Because MPC SkmKO mice showed a marked resistance to fat mass gains and increased insulin sensitivity on a normal chow diet, we sought to understand whether disrupting muscle MPC activity would attenuate high fat diet-induced obesity and metabolic dysfunction. WT and MPC SkmKO littermate pairs were placed on high fat diet (HFD) and monitored over time for changes in weight and body composition (*Figure 6A*). Body weight, fat mass, and lean mass were measured twice weekly through the first 16 weeks on HFD and periodically up until age 49 weeks and were not different between genotypes (*Figure 6B, C and D*, *Figure 6—figure supplement 1*). At age 49 weeks mice were switched to a micronutrient-matched synthetic control normal fat diet with 10% fat (NFD). Body weight and composition were measured weekly following diet switch. MPC SkmKO mice lost more body fat but equally retained lean mass during weight loss (*Figures 6E, F and G*). Food intake was not different between WT and MPC SkmKO mice when measured during HFD and 2 weeks into NFD, when weight loss rate was greatest (*Figure 6—figure supplement 1*).

We performed fasting and refeeding experiments after 48 weeks of HFD feeding and again 14 weeks after switching to NFD feeding. At both time points MPC SkmKO mice had similar blood

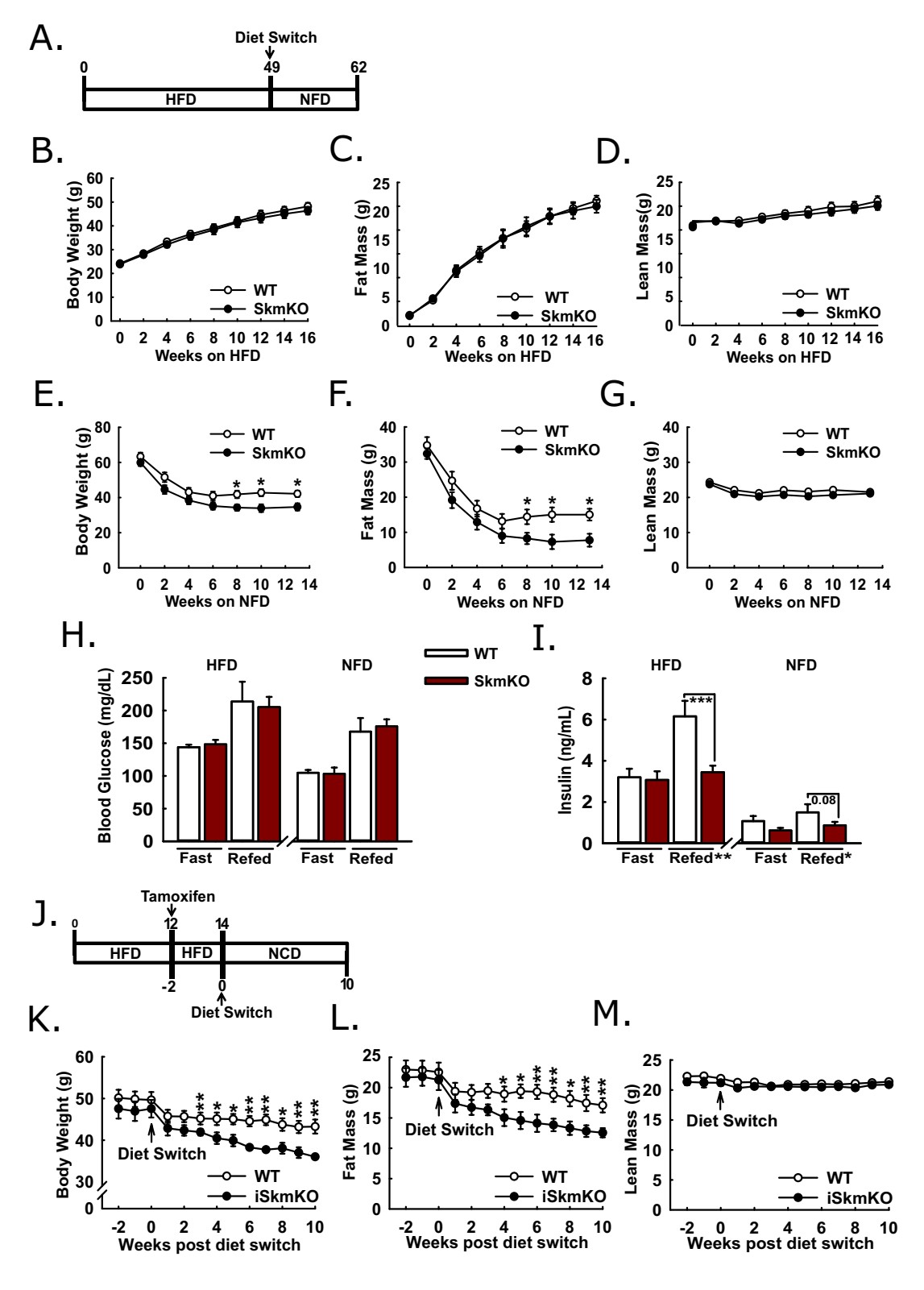

**Figure 6.** Protection and recovery from high fat diet-induced obesity. (A) Schema illustrating the time course of high fat diet (HFD) feeding and a switch to synthetic normal fat control diet (NFD). (B, C, and D) Body weight (B), fat mass (C), and lean mass (D) of WT and MPC SkmKO mice measured by NMR during HFD feeding of WT and MPC SkmKO mice (ages 9–25 weeks, n = 7–8, littermates, two-tailed t-test). (E, F, and G) Body weight (E), fat mass (F), and lean mass (G) of WT and MPC SkmKO mice measured by NMR during post-HFD, normal fat synthetic control diet (NFD) feeding of WT

*Figure 6 continued on next page*

*Figure 6 continued*

and MPC SkmKO mice (ages 58–71 weeks, n = 5–6, littermates, two-tailed t-test). (H, I) Fasted and refed blood glucose (H) and serum insulin (I) levels at ends of HFD and post-HFD, NFD treatments of WT and MPC SkmKO mice (ages 58 and 71 weeks, n = 5–6, littermates, two way ANOVA). (J) Schema illustrating the time course of mouse HFD treatment, tamoxifen injection for acute muscle-specific *Mpc1* deletion (MPC iSkmKO), and switch to normal chow diet (NCD) feeding. (K, L, and M) Body weight (K), fat mass (L), and lean mass (M) measured by NMR from end of 5 days tamoxifen administration, during 2 weeks continued HFD, and during 10 weeks post-HFD, NCD feeding of WT and MPC iSkmKO mice (ages 22–34 weeks, n = 6–7, littermates, two-tailed t-test). Data are presented as mean ± SEM (*p<0.05, **p<0.01, ***p<0.001).
DOI: https://doi.org/10.7554/eLife.45873.017

The following source data and figure supplements are available for figure 6:

**Source data 1.** Serum Parameters of 12 hr fasted and 30 min post refed WT and MPC SkmKO mice after 48 weeks of HFD and after 14 weeks of NFD recovery.
DOI: https://doi.org/10.7554/eLife.45873.020
**Figure supplement 1.** WT and MPC SkmKO mouse physiological parameters during high fat diet (HFD) and after return to normal fat diet (NFD) feeding.
DOI: https://doi.org/10.7554/eLife.45873.018
**Figure supplement 2.** Physiological parameters in MPC iSkmKO mice.
DOI: https://doi.org/10.7554/eLife.45873.019

glucose but lower serum insulin levels after refeeding. (*Figure 6H and I*). Serum triglyceride levels were lower in MPC SkmKO mice after fasting at the end of the HFD treatment and after both fasting and refeeding after NFD treatment (*Figure 6—source data 1*). These results demonstrate that constitutive muscle MPC disruption does not prevent HFD-induced obesity but protects against postprandial hyperinsulinemia during chronic high fat feeding and leads to more rapid onset of leanness when obese mice are returned to NFD.

To bypass potential development confounds that could occur with constitutive muscle MPC deletion model, we employed a tamoxifen-inducible Cre model. We generated tamoxifen-inducible MPC skeletal muscle specific knockout mice (iSkmKO) by crossing *Mpc1^fl/fl^* mice with mice expressing Cre fused to two modified estrogen receptor elements (mER-Cre-mER), driven by the human skeletal actin promoter (HSA) (*McCarthy et al., 2012*). Litter-mate WT and MPC iSkmKO mice were placed on 60% HFD for 12 weeks from age 10 to 22 weeks, treated with tamoxifen for acute MPC deletion, and two weeks later switched to a normal chow diet (NCD) (*Figure 6J*). Western blots of skeletal muscles at the end of the study confirmed MPC deletion (*Figure 6—figure supplement 2*). Compared to WT mice, MPC iSkmKO mice lost more body weight, which was almost exclusively from fat mass (*Figure 6K and L*). As with constitutive MPC SkmKO mice, iSkmKO mice retained lean mass equal to WT mice (*Figure 6M*). Body composition measurements were corroborated by adipose tissue and muscle weights (*Figure 6—figure supplement 2*). No food intake differences were observed between WT and MPC SkmKO mice during normal chow diet feeding after acute deletion (*Figure 6—figure supplement 2*). These results parallel and validate those from constitutive MPC SkmKO mice and demonstrate that enhanced recovery from obesity after diet switch does not require developmental preconditioning and may be elicited by acute MPC disruption.

## Discussion

How cells and organisms utilize pyruvate profoundly affects energy status, redox balance, and biosynthetic capacity (*Gray et al., 2014*). Pyruvate metabolism has been investigated extensively through studying the crucial regulators of mitochondrial pyruvate partitioning, pyruvate carboxylase and PDH. However, fundamental questions on the relationship between mitochondrial pyruvate utilization and disease have been challenging to address through their study alone. Here, we generated skeletal muscle-specific MPC knockout mice to test how decreasing muscle mitochondrial pyruvate uptake affects muscle and systemic metabolism.

After uptake by skeletal muscle, glucose may be channeled to three major fates: complete mitochondrial oxidation via pyruvate, storage as glycogen, or excretion as lactate. During hyperinsulinemic-euglycemic clamps, the preferred experiment for measuring insulin sensitivity in vivo, MPC disruption increased skeletal muscle insulin sensitivity and glucose uptake. In addition, under basal conditions, hepatic glucose production and peripheral glucose disposal were simultaneously

increased, indicating increased Cori Cycling. Importantly, this suggests that muscle glucose uptake and glycolysis are not intrinsically connected to pyruvate oxidation and that lactate excretion functions as flexible offset for balancing glucose uptake with disposal. This is notable given the role of lactate as a systemic energy currency (*Brooks, 2018*), including recent findings showing that most tissues other than brain, heart, and muscle utilize circulating lactate instead of glucose as their major source of pyruvate for mitochondrial oxidation (*Hui et al., 2017*). Thus, our findings demonstrate a nodal role for the muscle MPC in total-body energy supply by controlling circulating lactate fluxes.

Notably, the MPC SkmKO mouse leanness we observe is consistent with the unconventional view that elevated Cori Cycling in T2D is adaptive rather than maladaptive. Cori Cycle activity is increased in T2D and characterized as driving hyperglycemia (*Vaag et al., 1995*; *Zawadzki et al., 1988*). While this is accurate when considering the gluconeogenic component in isolation, we note that the full Cori Cycle does not produce net glucose. Muscle glycolysis is increased in T2D, which requires matching glucose uptake (*Giebelstein et al., 2012*; *Simoneau and Kelley, 1997*; *Vaag et al., 1995*). The Cori Cycle is futile because the two ATP molecules produced by glycolysis are traded for the six ATP equivalents required for re-synthesis of glucose from pyruvate (*Figure 7*). Surprisingly, while the energy-wasting nature of the Cori Cycle is addressed as a problem during cachexia and

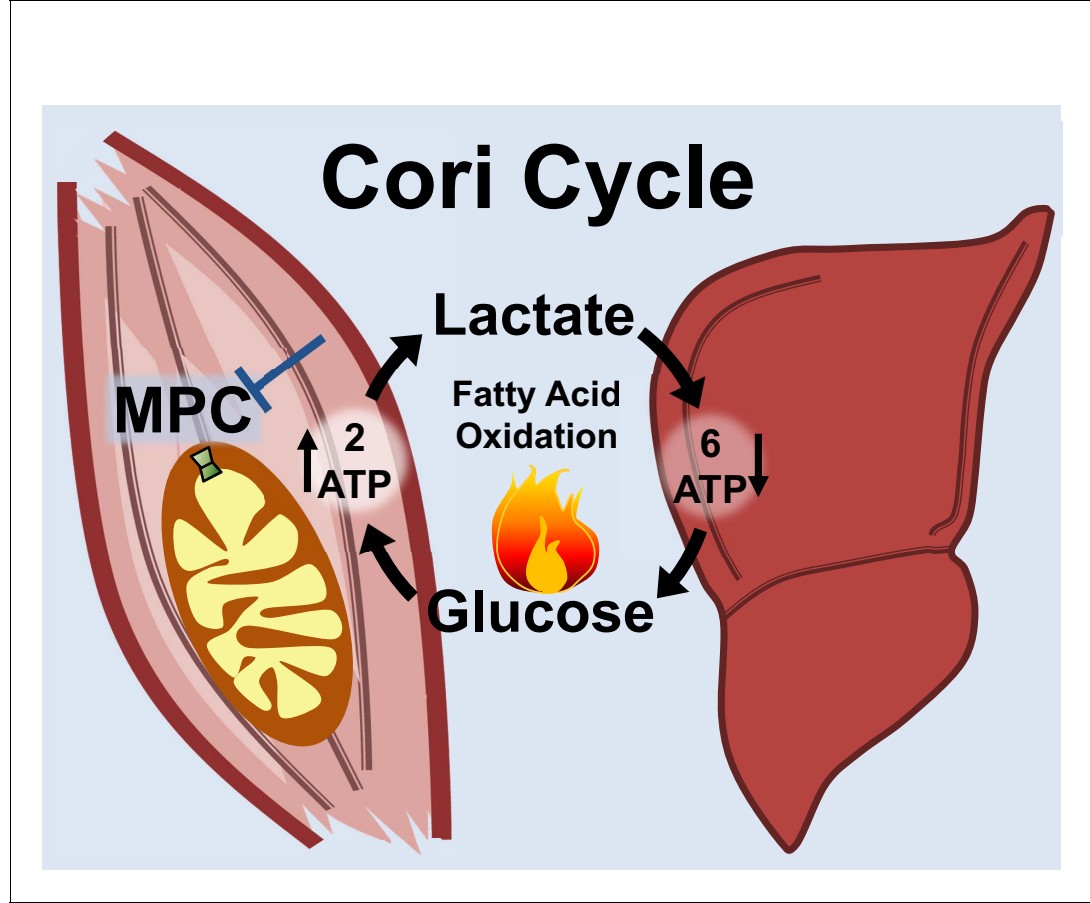

**Figure 7.** Skeletal muscle MPC disruption drives Cori cycling and fatty acid oxidation. Skeletal muscle MPC disruption (MPC SkmKO) impairs glycolytically generated pyruvate entry into skeletal muscle mitochondria, thereby increasing conversion of pyruvate to lactate and consequent skeletal muscle lactate excretion. Increased skeletal muscle lactate excretion drives hepatic gluconeogenesis that re-supplies glucose to skeletal muscle. Thus, skeletal muscle MPC disruption increases Cori Cycling. The Cori Cycle is energetically futile because each round produces two skeletal muscle ATP molecules and consumes six liver ATP equivalents, for a net whole-body consumption of 4 ATP equivalents. Because hepatic gluconeogenesis is energetically supported by fatty acid oxidation and muscle MPC disruption increases muscle fatty acid oxidation, futile Cori Cycling is energetically supported by fatty acid oxidation. Together, increased energy expenditure and fatty acid oxidation contribute to leanness arising from skeletal muscle MPC disruption.
DOI: https://doi.org/10.7554/eLife.45873.021

hyperthyroid states, it is virtually unaddressed in the diabetes and obesity literature (*Huang and Lardy, 1981*; *Potenza et al., 2009*; *Tisdale, 2009*). All other things being equal, elevated Cori Cycling helps solve the T2D metabolic conundrum of how to scavenge circulating glucose while clearing ectopic lipid deposits by oxidation. Because obesity often arises from a minimal daily energy surplus accumulated over time, we argue that small metabolic rate increases caused by elevated Cori Cycling can nonetheless impact long-term body composition.

Lastly, we consider limitations and potential future directions. The physiological changes occurring with skeletal muscle MPC disruption are clearly complex. Accounting for the many mechanisms that could be driving MPC SkmKO mouse leanness is a formidable challenge. By definition, because of the muscle-liver ATP exchange energetics, Cori Cycling increases whole-body energy expenditure. Additional factors are likely at play as well. These could include altered hypothalamic regulation by unidentified muscle-released serum factors, changes in intramuscular calcium cycling, which is ATP-intensive and known to be intricately connected with metabolism, and yet to be discovered lipolytic cross-talk between muscle and adipose tissue.

Though the feasibility of therapeutically targeting the MPC in vivo in humans has yet to be fully tested, we note that PPARγ-sparing thiazolidinedione-like molecules that inhibit the MPC attenuate diabetes in rodents and are now being tested in humans (*Colca et al., 2018*; *McCommis et al., 2015*; *McCommis et al., 2017*). Because pyruvate-alanine cycling can bypass the MPC to partially sustain glucose oxidation, in vivo targeting may therapeutically modulate rather than pathologically constrain mitochondrial metabolism. Nonetheless, inactivating human patient *MPC1* mutations impair neurological development leading to lasting health problems and in the case of one patient early death (*Bricker et al., 2012*; *Brivet et al., 2003*; *Oonthonpan et al., 2019*). How in vivo pharmacologic MPC modulation affects the function of tissues less metabolically plastic than liver and skeletal muscle, like brain, heart, and kidney, is a critical consideration that will need to be fully addressed. Important future work also includes deciphering the mechanisms by which skeletal muscle MPC disruption increases glucose uptake. This will require carefully interrogating insulin and AMPK signaling, glucose transporter regulation, and glycogen metabolism. In conclusion, our findings here raise the possibility that selectively decreasing skeletal muscle pyruvate uptake in obese and T2D patients may aid fat mass loss and restoration of whole-body insulin sensitivity.

# Materials and methods

## Key resources table

| Reagent type (species) or resource | Designation | Source or reference | Identifiers | Additional information |
|---|---|---|---|---|
| Genetic reagent (*Mus musculus*) | Mpc1$^{flox/flox}$, C57BL/6J | *Gray et al., 2015* | | |
| Genetic reagent (*Mus musculus*) | Myogenin-Cre, C57BL/6J | *Li et al., 2005* | | Gift from Dr. Eric Olsen |
| Genetic reagent (*Mus musculus*) | HSA-MerCreMer, C57BL/6J | *McCarthy et al., 2012* | | Gift from Dr. Karyn Esser |
| Antibody | Rabbit monoclonal anti-MPC1 | Proteintech | S4154-2 | Gift from Dr. Brian Finck; (1:1000) |
| Antibody | Rabbit monoclonal anti-MPC2 (D4I7G) | Cell Signaling Technology | #46141, RRID:AB_2799295 | (1:1000) |
| Antibody | Rabbit monoclonal anti-VDAC (D73D12) | Cell Signaling Technology | #4661, RRID:AB_10557420 | (1:1000) |
| Antibody | Mouse monoclonal anti-Actin (AC-15) | Sigma | #A5441, RRID:AB_476744 | (1:10000) |
| Antibody | Rabbit monoclonal anti-Tubulin (DM1A) | Cell Signaling Technology | #3873S, RRID:AB_1904178 | (1:1000) |

*Continued on next page*

*Continued*

| Reagent type (species) or resource | Designation | Source or reference | Identifiers | Additional information |
|---|---|---|---|---|
| Antibody | Rabbit monoclonal anti-HSP90 | Cell Signaling | #4874, RRID:AB_2121214 | (1:1000) |
| Antibody | Total OXPHOS cocktail | Abcam | ab110413, RRID:AB_2629281 | (1:1000) |
| Antibody | Mouse monoclonal anti-AMPKα (F6) | Cell Signaling Technology | #2793, RRID:AB_915794 | (1:1000) |
| Antibody | Rabbit monoclonal anti-pAMPKα (Thr172) (40H9) | Cell Signaling Technology | #2535, RRID:AB_331250 | (1:1000) |
| Antibody | Rabbit monoclonal anti-AKT (pan) (11E7) | Cell Signaling Technology | #4685, RRID:AB_2225340 | (1:1000) |
| Antibody | Rabbit polyclonal anti-pAKT (Ser473) | Cell Signaling Technology | #9271, RRID:AB_329825 | (1:1000) |
| Antibody | Mouse monoclonal anti-PDH-E1α (D6) | Santa Cruz Biotechnology | #SC-377092, RRID:AB_2716767 | (1:1000) |
| Antibody | Rabbit polyclonal anti-pPDH-E1α (Ser232) | Millipore Sigma | #AP1063, RRID:AB_10616070 | (1:1000) |
| Antibody | Rabbit monoclonal anti-GAPDH (D16H11) | Cell Signaling Technology | #5174, RRID:AB_10622025 | (1:20000) |
| Antibody | Goat anti-Mouse Dylight 800 | ThermoFisher | SA5-10176, RRID:AB_2556756 | (1:10000) |
| Antibody | Donkey anti-Rabbit DyLight 680 | ThermoFisher | SA5-10042, RRID:AB_2556622 | (1:5000) |
| Antibody | Goat anti-Rabbit DyLight 800 | ThermoFisher | #35571, RRID:AB_614947 | (1:10000) |
| Sequence-based reagent | 36b4 | Forward: 5'-CG TCCTCGTTGGAGTGACA | Reverse: 5'-CGGTGC GTCAGGGATTG | |
| Sequence-based reagent | Mpc1 | Forward: 5'-AACTACGAGA TGAGTAAGCGGC | Reverse: 5'-GTGTTT TCCCTTCAGCACGAC | |
| Sequence-based reagent | Mpc2 | Forward: 5'-CCGCTTTAC AACCACCCGGCA | Reverse: 5'-CAGCACACACC AATCCCCATTTCA | |
| Sequence-based reagent | Cpt1b | Forward: 5'-GGTCCCATAA GAAACAAGACCTCC | Reverse: 5'-CAGAAAGTACC TCAGCCAGGAAAG | |
| Sequence-based reagent | Hadha | Forward: 5'-TGGATGTGG ATGACATTGCT | Reverse: 5'-GGGGAAGAG TATCGGCTAGG | |
| Sequence-based reagent | Echs1 | Forward: 5'-CTTCACTG TAAGGGCAGGTG | Reverse: 5'-CTTGAGTTGGG AATCAGCAG | |
| Commercial assay or kit | FGF21 ELISA kit | ThermoFisher | NC9903102 | |
| Commercial assay or kit | Glucose Assay Kit | Sigma-Aldrich | HK20 | |
| Commercial assay or kit | High-Capacity cDNA Reverse Transcription kit | Applied Biosystems | 4368814 | |
| Commercial assay or kit | Infinity Cholesterol Reagent | Thermo Scientific | TR13421 | |
| Commercial assay or kit | Infinity Triglyceride Reagent | Thermo Scientific | TR22421 | |
| Commercial assay or kit | Leptin ELISA kit | R and D Systems | MOB00 | |
| Commercial assay or kit | Serum Ketone Kit | Wako Diagnostics | 415–73301, 411–73401, 412–73791 | |

*Continued on next page*

Continued

| Reagent type (species) or resource | Designation | Source or reference | Identifiers | Additional information |
|---|---|---|---|---|
| Commercial assay or kit | Serum NEFA kit | Wako Diagnostics | 999–34691, 995–034791, 991–34891, 993–35191, 276–76491 | |
| Commercial assay or kit | Ultra-sensitive Mouse Insulin ELISA kit | Crystal Chem | 90080 | |
| Commercial assay or kit | Insulin chemiluminescence ELISA | American Laboroatory Products | #80-INSMR-CH01 | |
| Chemical compound, drug | [1–14C]−2-deoxy-D-glucose | Perkin Elmer | NEC495001MCSBF3 | |
| Chemical compound, drug | α-Cyano-4-hydroxycinnamic acid (CHC) | Sigma-Aldrich | 476870 | |
| Chemical compound, drug | $^{13}$C3-Sodium Pyruvate (99%) | Cambridge Isotope Laboratories | 142014-11-17 | |
| Chemical compound, drug | $^{13}$C3-Sodium-L-Lactate (98%) | Cambridge Isotope Laboratories | CLM-1579 | |
| Chemical compound, drug | $^{14}$C-Palmitic acid | Perkin Elmer | NEC075H250UC | |
| Chemical compound, drug | $^{14}$C-Sodium Pyruvate | Perkin Elmer | NEC256050UC | |
| Chemical compound, drug | Antimycin A | Sigma-Aldrich | A8674 | |
| Chemical compound, drug | D-[3-$^3$H]-glucose | Perkin Elmer | NET331C001MC SBF3 | |
| Chemical compound, drug | Insulin | Novo Nordisk | Novolin R | |
| Chemical compound, drug | Deoxy-D-glucose 2-[1,2-$^3$H(N)] | Perkin Elmer | NET328A001MC | |
| Chemical compound, drug | Ensure | Abbott | Vanilla-57g | |
| Chemical compound, drug | FCCP | Sigma-Aldrich | C2920 | |
| Chemical compound, drug | High Fat Diet 60%Kcal from fat (HFD) | Research Diets Inc | D12492 | |
| Chemical compound, drug | Low Fat Diet 10% Kcal from fat (control diet, NFD) | Research Diets Inc | D12450J | |
| Chemical compound, drug | Normal Chow/Teklad Global Soy Protein-Free Extruded Rodent Diet Irradiated (NCD) | Envigo | 2920X | |
| Chemical compound, drug | Oligomycin A | Sigma-Aldrich | 75351–5 MG | |
| Chemical compound, drug | Rotenone | Sigma-Aldrich | R8875-1G | |
| Chemical compound, drug | Tamoxifen | Sigma-Aldrich | T5648-1G | |
| Chemical compound, drug | Triolein, [9,10-3H(N)] | Perkin Elmer | NET431001MC | |
| Chemical compound, drug | UK-5099 | ThermoFisher | 418610 | |
| Chemical compound, drug | Acetyl coenzyme A sodium salt | Sigma-Aldrich | A2056-10MG | |

*Continued*

| Reagent type (species) or resource | Designation | Source or reference | Identifiers | Additional information |
|---|---|---|---|---|
| Chemical compound, drug | $^{3}$H-Acetyl-CoA | Perkin Elmer | NET290050UC | |
| Chemical compound, drug | Malonyl coenzyme A lithium salt | Sigma-Aldrich | M4263-5MG | |
| Chemical compound, drug | Purified chicken fatty acid synthase | David Thomson Lab, BYU | | |
| Chemical compound, drug | Protease Arrest | G Biosciences | 786–437 | |
| Other | Accupulser | World Precision Instrument | A310 Accupulser | |
| Other | Body Composition Analyzer | Bruker | LF50-BCA Analyzer | |
| Other | Callipers | MSC Industrial Supply | 35518166 | |
| Other | Glucose Meter | LifeScan | Onetouch ulta mini | |
| Other | Glucose Strips | Diabetic Express | Feb-67 | |
| Other | Headstage | Axon Instruments | CV 203BV | |
| Other | Integrating patch clamp | Axon Instruments | Axopatch 200B | |
| Other | Lactate Meter | Nova Biomedical | Lactate plus meter | |
| Other | Lactate Strips | Nova Biomedical | NC0071872 | |
| Other | Low noise data acquisition system | Axon Instruments | Axon digidata 1550 | |
| Other | Low noise data acquisition system | Axon Instruments | Axon digidata 1440A | |
| Other | Microvette blood collection tubes | Sarstedt Inc | NC9141704 | |
| Other | Ohaus Hand-Held Scales | ThermoFisher | S65222 | |
| Other | Promethion Cages | Sable Systems International | Promethion Line | |
| Other | Rodent Treadmill | Columbus Instruments | Exer 3/6 Treadmill | |
| Other | Stimulus Isolater | World Precision Instrument | Stimulus Isolater | |
| Other | Western Blot Imager | Li-Cor | Odyssey CLx | |
| Other | Platform | Aurora Scientific | 809B | |
| Other | Force tranducer | Aurora Scientific | 305C | |
| Other | Refridgerated/Heated bath circulator | Thermo Fisher | 6200 R20F | |
| Other | XF96 4-port FluxPak with PET Microplates | Seahorse Bioscience | 102416–100 | |
| Software/tool | Sigmaplot | Sigmaplot | RRID:SCR_003210 | |
| Software/tool | Excel | Microsoft | RRID:SCR_016137 | |

## Animal studies

Animal work was performed in accord with the University of Iowa Animal Use and Care Committee (IACUC). *Mpc1*$^{fl/fl}$ (*Gray et al., 2015*), Myogenin-Cre (*Li et al., 2005*), HSA-MerCreMer (*McCarthy et al., 2012*) mice were maintained on a C57BL/6J genetic background. For all experiments, *Mpc1*$^{fl/fl}$ mice were backcrossed to C57BL/6J for six generations and male littermate paired mice were used for experiments unless otherwise indicated. Mouse ages and experimental sample

size including the number of littermates is indicated in figure legends. Mice were fed either chow, a high-fat diet (HFD) consisting of 60% kcal from fat, or a corresponding 10% kcal from fat control diet corresponding to the HFD (NFD). For studies utilizing the HSA-MerCreMer mice, recombination of the $Mpc^{fl/fl}$ allele was accomplished by intraperitoneal injection of 75 mg/kg tamoxifen for five consecutive days. Body composition was measured using a rodent-sized NMR machine (Bruker Minispec LF50).

## Western blot analysis

Snap-frozen tissues were pulverized (only for skeletal muscle and brown adipose tissue) and homogenized on ice in a lysis buffer containing 40 mM HEPES, 120 mM NaCl, 50 mM NaF, 5 mM sodium pyrophosphate, 5 mM β-glycerophosphate, 1 mM EDTA, 1 mM EGTA, 10% (v/v) Glycerol, 1% (v/v) Igepal CA-630, 1 μM DTT, and protease inhibitor. Samples were rotated at 4°C for 30 min followed by centrifugation at 21000 x $g$ at 4°C for 15 min and supernatant was collected. Protein concentration was determined by Bradford assay and an appropriate volume of 4X Laemmli buffer was added and all samples were incubated at 100°C for 10 min.

An equal amount of protein per sample was resolved using SDS-Glycine PAGE or SDS-Tricine PAGE (for Mpc1 and Mpc2). Proteins were transferred to a 0.22 μm nitrocellulose membrane before being probed with the specified antibodies. Primary antibody information: Mpc1 (1:1000, Proteintech), Mpc2 (1:1000, Cell Signaling Technology #46141), VDAC (1:1000, Cell Signaling Technology #4661), Actin (1:10,000, Sigma A5441), Tubulin (1:1000, Cell Signaling Technology #3873S), Total OXPHOS (1:1000, Abcam ab110413), HSP90 (1:1000, Cell Signaling Technology #4874), Total AMPK (1:1000, Cell Signaling Technology #2793), pAMPK T172 (1:1000, Cell Signaling Technology #2535), AKT (1:1000, Cell Signaling Technology #4685), pAKT S473 (1:1000, Cell Signaling Technology #9271), Total PDH-E1 (1:1000, Santa Cruz Biotechnology #SC-377092), pPDH-E1 S232 (1:1000, Millipore Sigma #AP1063), and GAPDH (1:20000, Cell Signaling Technology #5174). Secondary antibody information: Goat anti-Mouse DyLight 800 (1:10,000, ThermoFischer SA5-10176), Donkey anti-Rabbit DyLight 680 (1:5,000, ThermoFischer SA5-10042), Goat anti-Rabbit DyLight 800 (1:10,000, ThermoFischer 35571).

## Glucose and lactate/pyruvate tolerance tests

Mice were individually housed and fasted for 6 hr. To begin the experiment, blood glucose and lactate were measured using a One Touch UltraMini glucometer and Nova Biomedical Lactate Plus lactate meter via the tail vein. Mice were then intraperitoneally injected with a 10% glucose solution for GTT (2.0 g/kg lean mass) or a 10% lactate/pyruvate (10:1) solution for L/PTT (3.0 g/kg lean mass). Blood glucose and lactate were measured at 15, 30, 60, 90 and 120 min post-injection.

## Exercise tolerance test

Mice were acclimatized to the treadmills by two bouts of 30 min of 5 m/min walking on consecutive days. The exercise tolerance test (ETT) was performed on the third day by beginning at 4 m/min for 30 mins then increasing the treadmill speed by 2 m/min every 10 min for 90 min. After 90 min speed was increased by 4 m/min every 10 min until exhaustion was reached. Exhaustion was determined by inability of the mice to leave the treadmill shock grids for 10 consecutive seconds. Shock grids were set to irritate but not harm the mice.

## Skeletal muscle mitochondrial isolation

Skeletal muscle mitochondrial isolation was performed essentially as described previously (*Garcia-Cazarin et al., 2011*). Briefly, mouse gastrocnemius, soleus, tibialis anterior, extensor digitorum longus, and hamstring muscles were dissected, minced and incubated in 5 ml of PBS containing 10 mM EDTA and 0.01% (v/v) trypsin for 30 min. Following incubation, minced muscle was separated from digestion media and homogenized using a Teflon on glass dounce homogenizer in a buffer containing 10 mM EDTA, 215 mM D-mannitol, 75 mM Sucrose, 0.1% (w/v) fatty acid-free bovine serum albumin (BSA), and 20 mM HEPES pH 7.4. Homogenates were centrifuged at 700 x $g$ for 10 min at 4°C and the supernatant were collected. These supernatants were centrifuged at 10,500 x $g$ for 10 min at 4°C, and the resulting pellet was used as the mitochondrial fraction.

## Pyruvate uptake assay

The pyruvate uptake protocol was based on previously published methodology (*Gray et al., 2016*). Mitochondria were resuspended to 5.0–9.0 mg/mL in Uptake Buffer (120 mM KCL, 5 mM $KH_2PO_4$, 1 mM EGTA, 5 mM HEPES pH 7.4, 1 µM Rotenone, and 1 µM Antimycin A) and were divided into two equal aliquots treated with 2 mM α-Cyano-4-hydroxycinnamic acid (CHC) or vehicle. 20 µL of treated mitochondria were rapidly mixed with 20 µL of 2X Pyruvate buffer (Uptake Buffer, pH 6.2 with 0.10 mM $^{14}$C-Pyruvate) generating the pH gradient needed to initiate uptake. After 1 min, 80 µL of Stop buffer (Uptake Buffer pH 6.8 supplemented with 10 mM CHC) was rapidly mixed with the samples to halt uptake. Mitochondria were recovered by passing the solution through dual filter system consisting of a 0.8 µM cellulose filter and a 0.45 µM nitrocellulose filter. Filters were washed twice with 200 µL Wash buffer (Uptake Buffer pH 6.8 supplemented with 2 mM CHC and 10 mM Pyruvate). Excess filter material around the separated and washed mitochondria were removed, and filters containing mitochondria were placed into scintillation vials for quantification. Mitochondria pre-treated with CHC were used as a negative control and counts were subtracted from non-pre-treated mitochondria. Samples were normalized to the amount of mitochondrial protein used.

## Oxygen consumption rate (OCR) measurements

A Seahorse Bioscience XF-96 extracellular flux analyzer was used to monitor mitochondrial oxygen consumption similar to how previously described (*Rogers et al., 2011*). 5 µg of isolated skeletal muscle mitochondria suspended in a buffer containing 70 mM Sucrose, 220 mM D-mannitol, 10 mM $KH_2PO_4$, 5 mM $MgCl_2$, 5 mM HEPES pH 7.2, 1 mM EGTA, and 0.2% fatty acid free BSA were attached to V3-PET seahorse plates by centrifugation at 2000 x *g* for 20 min. Substrates of interest were at final concentrations of 10 mM Pyruvate/1 mM Malate or 10 mM Glutamate/1 mM Malate. A three injection protocol was utilized with three replicate measurements taken between each injection. Each replicate consisted of a 1 min mix step, a 1 min equilibration step, and a 3 min measurement step. After basal measurements were acquired, maximum oxygen consumption was stimulated by the addition of 4 mM ADP and 1 µM FCCP (Port A injection). MPC specific activity was inhibited by the addition of 1 mM CHC (Port B injection). Finally, Complex I activity was inhibited by 5 µM Rotenone (Port C injection). Oxygen consumption was normalized to mitochondrial protein loading.

## Fatty acid uptake assay

4 hr fasted mice were anesthetized with isoflurane and retro-orbitally injected with 200 µL 0.5% $^3$H-Intralipid. Fatty acid uptake was performed as previously described (*Bartelt et al., 2011*; *Cushing et al., 2017*; *Kusminski et al., 2012*). Briefly, blood samples were taken via the tail vein at 1, 5, 10, and 15 min after injection. After the last blood draw, the mice were anesthetized with isoflurane, and tissues were harvested and weighed. Approximately 50 mg of each tissue was then weighed and placed in 2:1 chloroform:methanol overnight at 4°C. 1 mL of 2 M $CaCl_2$ was then added to each sample to separate organic and aqueous layers. The samples were centrifuged for 10 min at 1500 rpm, and the upper aqueous layer was collected. The lower organic layer was evaporated overnight to remove chloroform. Blood samples, the upper aqueous layer, and the remaining organic layer following overnight evaporation were mixed with scintillation fluid and $^3$H counts per minute (CPMs) was measured by scintillation counting.

## Serum analysis

Mouse blood was collected via the tail vein into microvettes (Sarstedt) and spun at 3000 rpm for 30 min 4°C to separate serum. Insulin measurements were performed using the Ultra-Sensitive Mouse Insulin ELISA Kit (Crystal Chem). Serum cholesterol and triglycerides levels were determined using the Infinity Cholesterol and Infinity Triglyceride reagents (Thermo). Serum ketone and NEFA levels were measured using the Total Ketone Bodies and NEFA Reagents (Wake). All reagents were used according to manufacturer's directions.

## Ex vivo fatty acid oxidation

Tendon-to-tendon EDL muscle was carefully dissected from 13 week old mice, weighed (25–29 g), and placed in ice cold Krebs-HEPES-biocarbonate (KHB: 140 mM NaCl, 3.6 mM KCl, 0.5 mM $NaH_2PO_4$, 0.2 mM $MgSO_4$, 1.5 mM $CaCl_2$, 12.5 mM HEPES, 2 mM $NaHCO_3$, pH 7.4, gassed with

 

95%$O_2$/5%$CO_2$) containing 5 mM glucose and 1 mM ATP until assay initiation (30–60 min). Muscles were transferred to microcentrifuge tubes containing warmed (37°C) pre-reaction media (KHB, gassed with 95%$O_2$/5%$CO_2$, pH = 7.4, 12.5 mM HEPES, 1 mM L-carnitine, 200 µM palmitate conjugated to 0.25% fatty acid free BSA, 5 mM glucose, and 0.2% ethanol [palmitate vehicle]) for 15 min. Muscles were then placed in reaction media (Pre-reaction media plus 0.1% 0.5 µCi/mL [1-[14]C]-palmitate) in an airtight chamber with an NaOH solution containing collection vessel and incubated for 1 hr. 70% perchloric acid was injected into the chamber to release $CO_2$ from the reaction media that was subsequently trapped in the collection vessel. The NaOH was collected to assess complete palmitate oxidation to $CO_2$ and reaction media was retained to assess incomplete palmitate oxidation intermediates as acid soluble metabolites using a scintillation counter. Data were normalized to tissue weight, corrected for background and reaction media counts, and expressed as nmol/g tissue/hr.

## Traced hyperinsulinemic-euglycemic clamps

A catheter was introduced into the jugular vein of mice. Mice were allowed to fully recover from surgery prior to clamp procedure. Clamps were performed in 5 hr fasted, unrestrained, conscious mice. Whole-body glucose flux was traced by infusion of D-[3-[3]H]-glucose. After an 80 min basal sampling period, insulin administration was initiated with a 25 mU bolus followed by 2.5 mU/kg/min continuous infusion. At 45 mins prior to clamp conclusion min, [1-[14]C]−2-deoxy-D -glucose was infused in a single bolus. [14]C-2 deoxy-D –glucose-6-phosphate tracer enrichment was used to measure glucose uptake into liver, heart, kidney, white adipose tissue, brown adipose tissue, and gastrocnemius. Glucose appearance and disappearance rates were calculated using Steele's equations (*Steele et al., 1956*). Plasma insulin was measured by chemiluminescence ELISA.

## Gene expression analysis

Total RNA from TA muscle was extracted using the TRIzol method. For quantitative real-time PCR (qPCR) analysis, an equal amount of RNA was reverse transcribed and qPCR reactions were carried out using SYBR Green mastermix. Relative abundance of mRNA was normalized to ribosomal protein 36B4.

## Muscle glycogen measurements

Muscle glycogen measurements were performed using the acid hydrolysis approach which has been described previously (*Passonneau and Lauderdale, 1974*). Briefly,~10 mg of tibialis anterior muscle was placed into tubes containing 250 µL 2 M HCl. The samples were boiled for 2 hr with intermittent vortexing. Samples were returned to the original 250 µL with water and neutralized with 250 µL 2 M NaOH and 10 µL 1M Tris, pH 7.4. Glucose concentrations were measured using a commercially available kit according to manufacturer's direction.

## Steady state metabolomics

Data were acquired from an Agilent 6545 UHD QTOF interfaced with an Agilent 1290 UHPLC. Metabolites were separated by using a Millipore Sigma SeQuant ZIC-pHILIC (150 mm x 2.1 mm, 5 µm) column. Solvents were A, 95% water in acetonitrile with 10 mM ammonium acetate and 5 µM phosphate, and B, 100% acetonitrile. A flow rate of 200 µL/min was applied with the following gradient (minutes, %B): 0, 94.7%; 2, 94.7%; 27, 36.8%; 35, 20.0%; 37, 20.0%; 39, 36.8%. For all experiments, 2 µL of metabolic extract was injected. MS parameters were as follows: gas, 200°C 4 L/min; nebulizer, 44 psi; sheath gas, 300°C 12 L/min; capillary, 3kV; fragmentor, 100V; scan rate, one scan/s. MS detection was carried out in both positive and negative modes with a mass range of 65–1,700 Da. Identifications were established by comparing the retention times and fragmentation data of compounds to model standards. All raw data files were converted into mzXML files by using msconvert. Data analysis was performed by using either Agilent's Profinder or in-house R packages.

## [13]C-Glucose tracing

6hr-fasted mice were intraperitoneally injected with 50% labeled 10% U[13]C-Glucose in water. Blood glucose and lactate were measured at time 0, 30, and 60 min. At 65 mins post injection, mice were anesthetized and a consistent liver lobe and TA muscles were freeze clamped at liquid nitrogen for

metabolomic analysis. Metabolomic analysis was performed as described previously (*Diakos et al., 2016*).

For metabolomics, tissue samples were transferred to a tube prefilled with 1.4 mm diameter ceramic beads and weighed. The bead tube was returned to the freezer and allowed to cool down to −80 ℃. Each tube received 900 µL extraction solution (methanol:water = 9:1, containing appropriate amount of internal standard D$^4$-succinate, kept at −20 ℃ and thoroughly mixed before use). The tissue was disrupted using an OMNI BeadRuptor (Omni International), at 6.45 mHz speed, for 30 s, followed by incubation at −20 ℃ for 1 hr. The tubes were centrifuged at 20,000 x *g* for 10 min at 4 ℃ to pellet the tissue debris. The obtained supernatant was collected, dried under vacuum prior to GC-MS analysis.

All GC-MS analysis was performed with a Waters GCT Premier mass spectrometer fitted with an Agilent 6890 gas chromatograph and a Gerstel MPS2 autosampler. Dried samples were suspended in 40 uL of a 40 mg/mL O-methoxylamine hydrochloride in pyridine and incubated for one hr at 30˚ C. 25 uL of N-methyl-N-trimethylsilyltrifluoracetamide was added and samples were incubated for 30 min at 37℃ with shaking. 1 uL of the sample was injected to the gas chromatograph inlet in the split mode at a 10:1 split ratio with the inlet temperature held at 250℃. The gas chromatograph had an initial temperature of 95℃ for one minute followed by a 40 ℃/min ramp to 110℃ and a hold time of 2 min. This was followed by a second 5 ℃/min ramp to 250℃, a third ramp to 350℃, then a final hold time of 3 min. A 30 m Restek Rxi-5 MS column with a 5 m long guard column was employed for chromatographic separation.

Data was collected using MassLynx 4.1 software (Waters). Targeted metabolite identification and quantification as a function of area under the curve for each peak was performed using QuanLynx (Waters). Data was normalized for extraction efficiency and analytical variation by mean centering the area of D$^4$-succinate. Metabolite identity was established using a combination of an in house metabolite library developed using pure purchased standards and the commercially available NIST library. Isotope incorporation was performed in the same manner as targeted analysis but measuring the area under the curve for each isotope and correcting for natural isotope abundance.

## Muscle malonyl-CoA measurements

Malonyl-CoA was determined as previously described (*McGarry et al., 1978*). Briefly, muscles from 4 hour-fast mice were homogenized in 6% perchloric acid, and neutralized. The neutralized extract was then incubated at 37℃ in a reaction mixture containing 13 nmoles/ml acetyl-CoA-H$^3$, NADPH, and purified chicken fatty acid synthase (FAS) for 60 mins. The reaction was stopped using 70% perchloric acid. The amount of H$^3$-fatty acid synthesized during the reaction was used to calculate the initial tissue content of malonyl-CoA in pmoles/mg tissue.

## 0.5 Hz and 1 Hz in situ contraction

In situ contraction was performed as previously described (*Zhu et al., 2014*). Briefly, anesthetized mice were placed on a platform (Aurora Scientific) heated by circulating water. The tibialis anterior (TA) was exposed through dissection of the skin. A 3–0 silk suture was tied around the patellar tendon and secured to the fixed pillar of the force transducer. A 3–0 silk suture was tied around the distal tendon and looped through the lever arm of the force transducer. Tyrode's solution warmed to 30℃ was dripped on the muscle. An Accupulser stimulator routed through a stimulus isolation unit was used to excite the peroneal nerve. The resting tension on the muscle was set by incrementally adjusting the controller lever arm, and thus muscle length (L0) and resting tension, and then delivering a supramaximal stimulus to the peroneal nerve until a maximal twitch force was obtained. The stimulation amplitude was then set by plotting a twitch force versus stimulus amplitude curve and then setting the stimulator output at two times the threshold for a maximal twitch. Isometric contraction was generated and data collected by using pCLAMP software (Molecular Devices) interfaced with a digital acquisition device. The other contralateral TA muscle was treated as the non-contracted, sham control. Twitch force was normalized to the calculated muscle cross-sectional area (mN/cm$^2$) using the muscle mass (in g) divided by the product of the presumed muscle density of 1.06 g/cm3 and the optimal fiber length (L0 ×0.6, in cm) and expressed as specific force (*Brooks and Faulkner, 1988*; *Hakim et al., 2011*). L0 was measured between suture knots on the muscle tendon and muscle mass was measured after excision. After 15 mins of contraction protocol,

contracted and sham TA muscles were rapidly excised, freeze clamped under liquid nitrogen and stored for metabolomics.

## Home cage measurements of free-living, whole-body metabolism

Mice were acclimated for 24 hrs and then monitored for 72 hrs in an environmentally controlled Promethion metabolic screening system fitted with indirect open circuit calorimetry, food consumption monitors, and activity monitors to measure activity and energy expenditure. Energy expenditure and respiratory exchange ratio (RER = $VCO_2/VO_2$) were calculated from the gas exchange data.

## Grip strength measurements

Grip strength measurements were performed as described previously (*Dyle et al., 2014*). Forelimb grip strength was determined using a grip strength meter equipped with a triangular pull bar (Columbus Instruments). Each mouse was subjected to five consecutive tests to obtain the peak value. The peak value was considered the maximum grip strength.

## Ex vivo muscle force production analysis

Briefly, Mice were euthanized and the lower hind limb was removed and immediately placed in Krebs Ringer solution with 95% O2% and 5% CO2. The gastrocnemius, soleus, TA muscles, as well as the distal half of the tibia and fibula, were removed, leaving the intact EDL muscle. Contractile measurements were made on the EDL muscle. Maximum isometric tetanic force were recorded along with muscle mass and optimal fiber length and normalized to cross-sectional area to calculate specific force.

## Quantification and statistical analysis

SigmaPlot or Microsoft Excel software suites were used to organize and statistically analyze data and prepare figures. Unless otherwise noted, data are represented as mean ± SEM, statistical significance was determined using a two-tailed Student's t-test or analysis of variance (ANOVA), and outliers were identified with the Grubbs' test.

## Acknowledgements

We thank the Fraternal Order of Eagles Diabetes Research Center Metabolic Phenotyping Core for assistance with insulin clamps. This work was supported by NIH grants R01 DK104998 and R00 AR059190 (EBT); R01 HD084645 and R01 HD082109 (RKS); R01 DK092412 (LVZ); R35ES028365 (GJP); R01 HL130146 (BSD); T32 HL007344 to Steven Lentz (RDS), F32 DK116522 (RDS), the Carver College of Medicine MSTP grant GM007337 (SCT); T32 HL007638 to Michael Welsh (AJR); ADA 1–18-PDF-060 (AJR); F32 DK101183 (LRG); T32 DK112751 to E Dale Abel (DAS); P30CA086862 to George Weiner, which contributed to support of core facilities utilized for this research.

## Additional information

### Funding

| Funder | Grant reference number | Author |
| --- | --- | --- |
| National Institutes of Health | DK104998 | Eric B Taylor |
| National Institutes of Health | AR059190 | Eric B Taylor |
| National Institutes of Health | HD084645 | Richard K Shields |
| National Institutes of Health | HD082109 | Richard K Shields |
| National Institutes of Health | DK092412 | Leonid V Zingman |
| National Institutes of Health | ES028365 | Gary Patti |
| National Institutes of Health | HL130146 | Brandon S Davies |
| National Institutes of Health | HL007344 | Ryan D Sheldon |
| National Institutes of Health | DK116522 | Ryan D Sheldon |

| National Institutes of Health | GM007337 | Sean C Tompkins |
| National Institutes of Health | HL007638 | Adam J Rauckhorst |
| National Institutes of Health | DK101183 | Lawrence R Gray |
| American Diabetes Association | 1-18-PDF-060 | Adam J Rauckhorst |
| National Institutes of Health | DK112751 | Diego A Scerbo |

The funders had no role in study design, data collection and interpretation, or the decision to submit the work for publication.

### Author contributions

Arpit Sharma, Ryan D Sheldon, Conceptualization, Data curation, Formal analysis, Investigation, Methodology, Writing—original draft, Writing—review and editing; Lalita Oonthonpan, Conceptualization, Data curation, Investigation; Adam J Rauckhorst, Data curation, Formal analysis, Investigation, Writing—original draft, Writing—review and editing; Zhiyong Zhu, Kevin Cho, Emily M Cushing, Data curation, Investigation; Sean C Tompkins, Lawrence R Gray, Alvin D Pewa, Investigation; Wojciech J Grzesik, Michael C Dyle, Data curation, Formal analysis, Investigation; Diego A Scerbo, Investigation, Writing—review and editing; James E Cox, Brandon S Davies, Data curation, Formal analysis, Investigation, Methodology; Chris Adams, Data curation, Investigation, Methodology; Richard K Shields, Funding acquisition, Investigation, Methodology; Andrew W Norris, Data curation, Formal analysis, Investigation, Methodology, Writing—original draft, Writing—review and editing; Gary Patti, Formal analysis, Investigation, Methodology; Leonid V Zingman, Data curation, Formal analysis, Investigation, Methodology, Writing—review and editing; Eric B Taylor, Conceptualization, Resources, Data curation, Formal analysis, Supervision, Funding acquisition, Validation, Investigation, Methodology, Writing—original draft, Project administration, Writing—review and editing

### Author ORCIDs

Arpit Sharma https://orcid.org/0000-0002-6132-1129
Lalita Oonthonpan https://orcid.org/0000-0001-9425-8448
Adam J Rauckhorst https://orcid.org/0000-0002-4101-0075
Emily M Cushing http://orcid.org/0000-0002-9495-802X
Gary Patti http://orcid.org/0000-0002-3748-6193
Eric B Taylor https://orcid.org/0000-0003-4549-6567

### Ethics

Animal experimentation: Animal work was performed in accordance with the University of Iowa Animal Use and Care Committee (IACUC). The University of Iowa IACUC is accredited by AALACi (#000833), is a Registered United States Department of Agriculture research facility (USDA No. 42-R-0004), and has PHS Approved Animal Welfare Assurance (#D16-00009).

### Decision letter and Author response

Decision letter https://doi.org/10.7554/eLife.45873.026
Author response https://doi.org/10.7554/eLife.45873.027

## Additional files

### Supplementary files

• Supplementary file 1. TA muscle metabolomic profiles after sham and in situ contraction. n = 5–6, littermates, age = 12 weeks, two way ANOVA.
DOI: https://doi.org/10.7554/eLife.45873.022

• Supplementary file 2. Percent isotopologue distribution of TCA cycle intermediates in muscle and liver 65 min after $U^{13}C$-labeled glucose injection of WT and MPC SkmKO mice. (n = 8, 7 littermates

one non-littermate, age = 12 weeks, two tailed t-test) Data are presented as mean ± SEM (*p<0.05, **p<0.01, ***p<0.001).
DOI: https://doi.org/10.7554/eLife.45873.023

• Transparent reporting form
DOI: https://doi.org/10.7554/eLife.45873.024

### Data availability

All metabolomic results generated as part of this study are provided in Supplemental tables 2 and 3 related to Figure 5.

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
