## [Decision Letter]

Thank you for submitting your article "Impaired skeletal muscle mitochondrial pyruvate uptake rewires glucose metabolism to drive whole-body leanness" for consideration by *eLife*. Your article has been reviewed by three peer reviewers, and the evaluation has been overseen by a guest Reviewing Editor and Mark McCarthy as the Senior Editor. The following individuals involved in review of your submission have agreed to reveal their identity: Darrell Neufer (Reviewer #1); Rachel Perry (Reviewer #2); Deb Muoio (Reviewer #3).

The reviewers have discussed the reviews with one another and the Reviewing Editor has drafted this decision to help you prepare a revised submission.

All reviewers and the Reviewing Editor found your work of significant interest. The observation that the mitochondrial pyruvate carrier can have such a profound role in regulating energy expenditure and thus body mass is of interest. However, all reviewers felt that the manuscript fell short in defining the mechanism for the reduced fat mass, leading to a concern that this will represent yet another paper that reports such a link without mechanism. Nevertheless, the reviewers felt it was of sufficient interest to allow you the chance of resubmitting your paper provided you can fully address the "essential" concerns. In summary they are:

First, one of the reviewers identified problems with the glucose clamp data. There appears to be no effect of insulin on peripheral glucose utilisation in the KO mice. This needs a complete explanation and you need to provide documentation of the insulin values achieved during the clamp.

Second, if muscle glucose uptake is increased, apparently in the absence of insulin elevation, this is deserving of further explanation. One reviewer requested measurement of AMPK activity in muscle from WT and KO mice ie assessment of phosphorylation of AMPK and phosphorylation of at least one AMPK substrate such as ACC. One of the reviewers also requested examination of PDH phosphorylation and so such studies could also be done in parallel with those of AMPK.

Third, it is distinctly possible that MPC deletion has caused adaptive changes in gene expression and these could also be explored in either a targeted or non-targeted manner.

Summary:

The study by Sharma investigates the metabolic impact of deletion of the mitochondrial pyruvate carrier (MPC) specifically in muscle using both a constitutive and inducible model. The studies appear to be technically sound and well executed, and the manuscript is generally well written. MPC has been knocked down/out in a number of contexts. The compensatory changes in fuel metabolism are pretty well established and/or predictable from those studies, which weakens the novelty of the study. The finding that MPC1 deletion specifically in muscle leads to reduced fat mass on a chow diet is interesting, but it is not clear nor explored experimentally why the same effect is not evident in mice on a chronic high fat diet. The increase in energy expenditure induced by the shift in fuel utilization is likely driving the phenotype, but the mechanism accounting for the increase in energy expenditure on a chow diet is not explored beyond speculation regarding the Cori Cycle. The findings regarding the accelerated weight loss and fat mass when SkmKO mice were switched from a chronic high fat diet to a normal chow diet are novel and interesting and the main strength of the paper. Limiting the ability of pyruvate to enter the mitochondria in muscle appears to induce a mild increase in energy expenditure that is protective in the context of normal chow diet, which raises the possibility of therapeutically targeting MPC to facilitate weigh loss.

Essential revisions:

1) The hyperinsulinemic-euglycemic clamps appear not to be of the highest quality. Specifically:

a) Basal glucose turnover rates (Ra) in a mouse are typically 10-15 mg/(kg-min); even in poorly-controlled diabetic mice, they rarely exceed 30 mg/(kg-min). The fact that Ra=40-60 mg/(kg-min) suggests that the mice may not have been at steady state. Please provide raw data (glucose and specific activity) to demonstrate that the mice are at steady state.

b) It does not appear that insulin stimulated whole-body glucose disposal (Figure 4D, comparing basal to steady state). In an insulin sensitive animal, insulin should markedly stimulate Rd. Please provide plasma insulin concentrations (basal and clamp) – are they matched? Is the lack of increase in Rd due to a relatively low steady state plasma insulin level?

c) Please provide GIR in mg/(kg-min) so that it can be compared to Ra and Rd.

2) The assays used to confirm that MPC1 KO results in decreased PDH flux and pyruvate oxidation were performed in isolated mitochondria (Figure 1). However, these assays do not capture transamination and/or carboxylation reactions occurring in the cytoplasm that could circumvent MPC1 to shuttle pyruvate-derived carbons into the TCAC. To this point, the data in Supplementary file 2 seem to suggest that under resting conditions, PDH flux (M+2) was similar between KO and controls – anaplerotic flux (M1 and M3) appeared lower in the KOs, and labeling of malate and lactate were unchanged. These results and their interpretation deserve some further discussion. Also, western blot analysis of PDH phosphorylation and labeling of alanine might be informative.

3) Investigators conclude that muscle insulin sensitivity is enhanced in the KO mice. It would be helpful to know if this phenotype is secondary to enhanced insulin signaling, or simply reflective of changes in glucose flux and partitioning. Investigators could address this by examining insulin signaling in existing specimens from the clamp experiments. Additionally, the overall findings seem to imply that muscle energy charge (ATP/ADP/AMP) might be compromised in KO mice. Therefore, an obvious question is whether or not MPC1 ablation leads to increased AMPK phosphorylation and activity, which would be expected to promote Glut4 translocation and glucose uptake. These questions are central to the take-home message and could be addressed with existing samples.

4) Investigators concluded that MPC1 KO caused increased fat oxidation due to diminished malonyl CoA levels and resulting inhibition of CPT1. However, the dramatic accumulation of even chain acylcarnitines (including long chain species listed in S. data table) in muscle of control mice suggests that CPT1 activity was not a limiting factor in this group. The authors should consider the possibility that enhanced fat oxidation results from adaptive upregulation of FAO enzymes (could be easily measured), and/or diminished substrate completion. The latter is supported by the low levels of C2 carnitine upon contraction, which probably indicates lower acetyl CoA levels.

---

## [Author Response]

Essential revisions:1) The hyperinsulinemic-euglycemic clamps appear not to be of the highest quality. Specifically:a) Basal glucose turnover rates (Ra) in a mouse are typically 10-15 mg/(kg-min); even in poorly-controlled diabetic mice, they rarely exceed 30 mg/(kg-min). The fact that Ra=40-60 mg/(kg-min) suggests that the mice may not have been at steady state. Please provide raw data (glucose and specific activity) to demonstrate that the mice are at steady state.b) It does not appear that insulin stimulated whole-body glucose disposal (Figure 4D, comparing basal to steady state). In an insulin sensitive animal, insulin should markedly stimulate Rd. Please provide plasma insulin concentrations (basal and clamp) – are they matched? Is the lack of increase in Rd due to a relatively low steady state plasma insulin level?c) Please provide GIR in mg/(kg-min) so that it can be compared to Ra and Rd.

We agree that extraordinarily high basal Ras and a lack of insulin-stimulated Rds were problematic. While the reasons for this are not certain, high basal Ra levels were observed in a limited and date-restricted series of clamps performed in our core – including the clamps under question. These extraordinarily high basal Ra levels resolved with the replacement of several under-delivering infusion pumps. In light of this coupled to the importance of clamp data for understanding whole-body glucose homeostasis, we repeated clamp experiments.

a) In the repeat clamps, basal Ra’s in mg/kg/min were 30 (WT) and 39 (MPC SkmKO) (Figure 4C). As expected, these values are decreased compared to the previous clamp experiments, with the controls ranging on the high end of literature-reported values. Importantly, the new clamps repeat the same key result of significantly higher basal Ra in MPC SkmKO mice compared to controls, consistent with elevated Cori Cycling.

b) In the new set of clamps, insulin administration (Figure 4—figure supplement 1) markedly stimulated Rd in both WT (30 to 70 mg/kg/min) and MPC SkmKO (39 to 87 mg/kg/min) mice (Figure 4D).

c) GIR is now provided in mg/kg/min for clear comparison to Ra and Rd.

2) The assays used to confirm that MPC1 KO results in decreased PDH flux and pyruvate oxidation were performed in isolated mitochondria (Figure 1). However, these assays do not capture transamination and/or carboxylation reactions occurring in the cytoplasm that could circumvent MPC1 to shuttle pyruvate-derived carbons into the TCAC. To this point, the data in Supplementary file 2 seem to suggest that under resting conditions, PDH flux (M+2) was similar between KO and controls – anaplerotic flux (M1 and M3) appeared lower in the KOs, and labeling of malate and lactate were unchanged. These results and their interpretation deserve some further discussion. Also, western blot analysis of PDH phosphorylation and labeling of alanine might be informative.

We now discuss the M+1 and M+3 vs M+2 isotopologue differences between WT and MPC SkmKO mice in the Results section, in the last paragraph of the “Adaptive Glutaminolysis and Pyruvate-Alanine Cycling” subsection. We added alanine labeling data to Supplementary file 2. Based on citrate and alanine isotopologue distribution differences between WT and MPC SkmKO mice, we discuss the possibility that pyruvate-alanine cycling more efficiently channels pyruvate through PDH for forward TCA cycle flux than through pyruvate carboxylase for anaplerosis. To understand whether increased PDH phosphorylation could be directing a greater MPC SkmKO muscle pyruvate proportion through forward vs anaplerotic TCA cycle flux, we performed new experiments measuring PDH phosphorylation in quadriceps muscle harvested at the end of clamp experiments (Figure 4—figure supplement 2). These blots show no differences between WT and MPC SkmKO mice PDH phosphorylation, consistent with other mechanisms controlling pyruvate channeling.

3) Investigators conclude that muscle insulin sensitivity is enhanced in the KO mice. It would be helpful to know if this phenotype is secondary to enhanced insulin signaling, or simply reflective of changes in glucose flux and partitioning. Investigators could address this by examining insulin signaling in existing specimens from the clamp experiments. Additionally, the overall findings seem to imply that muscle energy charge (ATP/ADP/AMP) might be compromised in KO mice. Therefore, an obvious question is whether or not MPC1 ablation leads to increased AMPK phosphorylation and activity, which would be expected to promote Glut4 translocation and glucose uptake. These questions are central to the take-home message and could be addressed with existing samples.

We agree that this is an interesting and important question. As recommended, we examined quadriceps muscle harvested at the end of clamp experiments by Western blot for Akt serine 473 and AMPK threonine 172 phosphorylation (Figure 4—figure supplement 2). Neither was increased in MPC SkmKO mice. We note in the manuscript that this doesn’t rule out that changes in either of these pathways contributed to increased MPC SkmKO glucose uptake, only that Akt and AMPK phosphorylation were not increased in post clamp samples. Deciphering the mechanisms accounting for increased MPC SkmKO muscle glucose uptake will likely be a complex and formidable challenge. Given the sizable experimental approach required to adequately address this question, we believe they are beyond the scope of this investigation. In the Discussion we raise this as an important future research direction.

4) Investigators concluded that MPC1 KO caused increased fat oxidation due to diminished malonyl CoA levels and resulting inhibition of CPT1. However, the dramatic accumulation of even chain acylcarnitines (including long chain species listed in S. data table) in muscle of control mice suggests that CPT1 activity was not a limiting factor in this group. The authors should consider the possibility that enhanced fat oxidation results from adaptive upregulation of FAO enzymes (could be easily measured), and/or diminished substrate completion. The latter is supported by the low levels of C2 carnitine upon contraction, which probably indicates lower acetyl CoA levels.

This is an excellent point of distinction. We now discuss the roles of both changed substrate competition and enzymes activities, based on new qPCR experiments, in increased MPC SkmKO fatty acid oxidation. Based on our data, we modified the results narrative to suggest that: 1) in resting MPC SkmKO muscle diminished substrate competition and decreased malonyl-CoA both contribute to increased fatty acid oxidation; and 2) in working (in situ contracted) MPC SkmKO muscle that diminished substrate competition predominates over Cpt1b disinhibition as a mechanism for increased fatty acid oxidation. As the reviewer notes, we discuss the relative accumulation of acetyl- and other even chain acyl carnitines with muscle contraction and how relatively lower acetyl-carnitine in MPC SkmKO muscle is consistent with diminished substrate competition for conversion to acetyl-CoA. Furthermore, we performed additional experiments measuring transcript levels for key fatty acid oxidation enzymes, *Cpt1b, Esch1*, and *Hadha* (Figure 5—figure supplement 2). We observed no differences between WT and MPC SkmKO mice, indicating the absence of programmatic upregulation of fatty acid oxidation gene expression as a mechanism for increased fatty acid oxidation.